# Safe Deep RL in 3D Environments using Human Feedback

## Abstract

Agents should avoid unsafe behaviour during both training and deployment. This typically requires a simulator and a procedural specification of unsafe behaviour. Unfortunately, a simulator is not always available, and procedurally specifying constraints can be difficult or impossible for many real-world tasks. A recently introduced technique, ReQueST, aims to solve this problem by learning a neural simulator of the environment from safe human trajectories, then using the learned simulator to efficiently learn a reward model from human feedback. However, it is yet unknown whether this approach is feasible in complex 3D environments with feedback obtained from real humans - whether sufficient pixel-based neural simulator quality can be achieved, and whether the human data requirements are viable in terms of both quantity and quality. In this paper we answer this question in the affirmative, using ReQueST to train an agent to perform a 3D first-person object collection task using data entirely from human contractors. We show that the resulting agent exhibits an order of magnitude reduction in unsafe behaviour compared to standard reinforcement learning.

## 1 Introduction

Many of deep reinforcement learning's recent successes have relied on the availability of a procedural reward function and a simulated environment for the task in question. As a result, research has been largely insulated from many of the difficulties of learning in the real world.

One of these issues is safe exploration (Garcıa & Fernández, 2015). Online reinforcement learning is dependent on first-hand experience in order to learn the constraints of safe behaviour: the agent must drive the car off a cliff to learn *not* to drive the car off a cliff. While such actions may be fine in simulation, in the real world these actions may have unacceptable consequences, such as injury to humans. Further, such constraints are not always easy to describe using procedural functions.

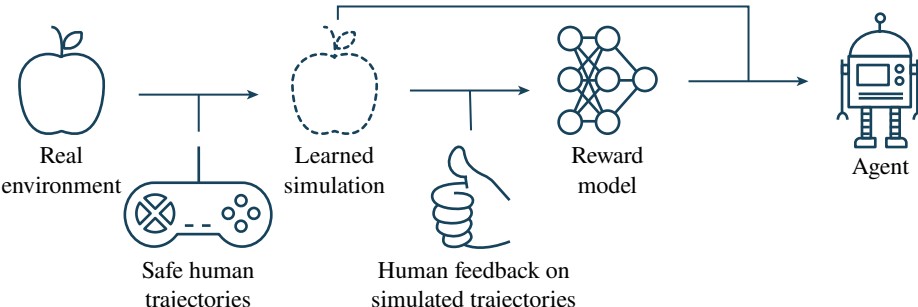

Figure 1: Our approach. Using a number of safe trajectories demonstrated by humans in the real environment, we train a dynamics model that functions as a *learned* simulator. Using this simulator, we train a reward model by asking humans for feedback on hypothetical trajectories. Once the reward model is robust, we deploy an agent in the real environment using model predictive control.

A recently proposed approach to safe exploration is *reward query synthesis via trajectory optimization*, ReQueST (Reddy et al., 2019). In this approach, the agent is trained in a *learned* dynamics model (a neural environment simulator) with rewards from a *learned* reward model. Given models of sufficient fidelity, this should allow us to train an agent with close to zero instances of unsafe behaviour in the real environment. However, as of Reddy et al. (2019), ReQueST has only been demonstrated to work in simple 2D environments – a simple navigation task, and a 2D car racing game – with a dynamics model learned from (potentially unsafe) random exploration, and a reward model learned from binary feedback generated by a procedural reward function.

In this work, we aim to answer the question: is ReQueST feasible in complex 3D environments, with data used to train both dynamics and reward models sourced from real humans? In particular, can we learn a pixel-based dynamics model of sufficient quality to enable informative human feedback, and are the data requirements viable, especially in terms of quantity? Our key contributions in this work are as follows.

- We demonstrate that ReQueST *is* feasible in a complex 3D environment, training a pixel-based dynamics model and reward model from 160 person-hours of safe human exploratory trajectories and 10 person-hours of reward sketches. We also show that performance degrades smoothly when models are trained on smaller amounts of data.
- On a 3D first-person object collection task, we show that ReQueST enables training of a competent agent with 3 to 20 times fewer instances of unsafe behaviour during training (close to *zero* instances if not counting mistakes by human contractors) than a traditional RL algorithm.

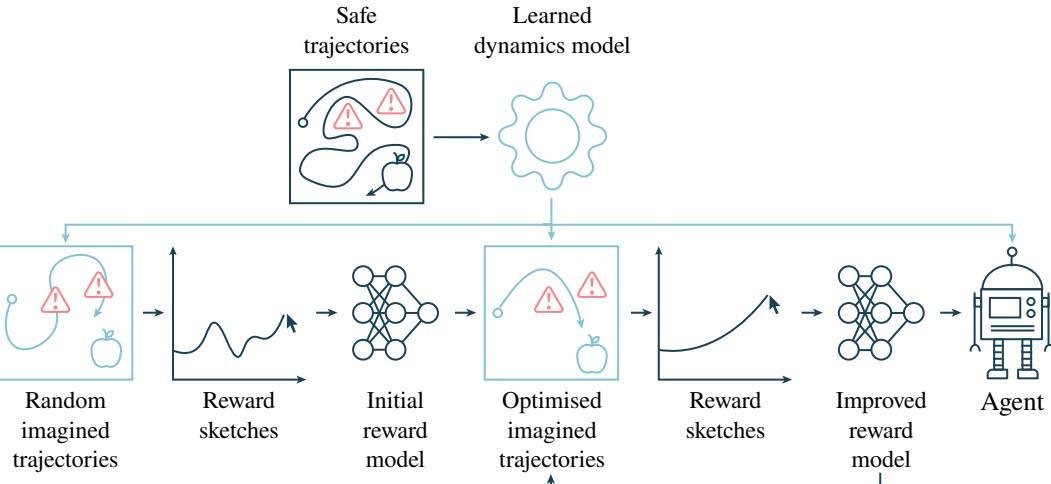

Figure 2: Our training procedure for an apple collection task. Light blue: steps performed using learned dynamics model. First, a human demonstrates a number of trajectories, exploring thoroughly while avoiding the unsafe (red) parts of the state space. From these trajectories, we train a dynamics model, and use it to generate a number of random trajectory videos. We ask humans to provide feedback on these videos in the form of reward sketches, then use these sketches to train an initial reward model. Since this reward model may not be sufficiently robust, we generate another set of trajectory videos by optimising for maximum and minimum reward as predicted by the current reward model, exposing for example instances where an agent would otherwise exploit the reward model. Using sketches on these trajectories, we train an improved reward model. This cycle can be repeated a number of times until a human deems the reward model to be good enough, at which point we deploy the agent using model predictive control.

## 2 RELATED WORK

**Safe exploration** Safe exploration has been studied extensively (García & Fernández, 2015). Most existing work achieves safety by making strong assumptions about the state space, such as all unsafe

states being known in advance (Geibel & Wysotzki, 2005; Luo & Ma, 2021) or the state space being reasonably smooth (Berkenkamp et al., 2017; Dalal et al., 2018). Other approaches require additional inputs, such as a procedural constraint function (Altman, 1999; Achiam et al., 2017; Ray et al., 2019; Dalal et al., 2018), a safe baseline policy (Garcia & Fernández, 2012), or a separate system that can determine whether an action is safe (Alshiekh et al., 2018). In contrast, the only assumption we make is that a *human* can recognise when a trajectory contains or is heading towards an unsafe state.

**Acceptability of unsafe behaviour**  Another important dimension is whether safety is treated as a *soft* or a *hard* constraint. Most work assumes the former, seeking to *minimise* time spent in unsafe states (e.g. Geibel & Wysotzki (2005)). Two notable examples of the latter include Saunders et al. (2017), which avoids unsafe behaviour by having a human intervene during training to block unsafe actions, and Luo & Ma (2021), which starts with a trivial-but-safe policy and slowly broadens the policy while guaranteeing the policy will avoid a set of unsafe states specified by the user in advance.

**Learned dynamics models in prior work**  The broad structure of our approach – learning a dynamics model (Chiappa et al., 2017) from trajectories, then learning a policy or planning using that model – has been successfully used in simple control tasks (Hafner et al., 2019), autonomous helicopter flight (Abbeel et al., 2010), fabric manipulation (Hoque et al., 2021), Atari games (Buesing et al., 2018), and in simple 3D environments (Ha & Schmidhuber, 2018). Our work shows that this approach is viable even with complex 3D scenes, with a dynamics model learned from human-demonstrated trajectories, and with a reward model learned from human feedback rather than relying on environment rewards (Hafner et al., 2019; Buesing et al., 2018), trajectory following (Abbeel et al., 2010), or maximisation of episode length (Ha & Schmidhuber, 2018).

**Prior work on ReQueST**  Our work expands on the original ReQueST (Reddy et al., 2019) in two main ways. First, we source all data from humans, showing that ReQueST is still applicable with imperfect, real-world data. Second, rather than simple 2D environments, we use visually-complex 3D environments, requiring a much more sophisticated dynamics model.

**Reward modeling**  As with the original ReQueST, we rely on a learned model of the reward function (Knox, 2012; Leike et al., 2018). In contrast to the classification-based reward model used in the original, our reward model regresses to a continuous-valued reward, trained using reward sketches (Cabi et al., 2019) on imagined trajectories. Other forms of feedback on which such reward models can be trained include real-time scalar rewards (Knox & Stone, 2009; Warnell et al., 2018), goal states (Bahdanau et al., 2018), trajectory demonstrations (Finn et al., 2016), trajectory comparisons (Christiano et al., 2017), and combinations thereof (Ibarz et al., 2018; Stiennon et al., 2020; Jeon et al., 2020).

# 3 METHODS

## 3.1 REQUEST

**Approach overview**  For online reinforcement learning algorithms, the only way an agent can learn about unsafe states is by visiting them. Intuitively, ReQueST avoids this problem by allowing agents to explore these states in a simulated model without having to visit them in the real environment. First, the agent learns a model of the environment by watching a human, who already knows how to navigate the environment safely. The agent then uses this model to 'imagine' various scenarios in the environment both safe and unsafe, asking the human for feedback on those scenarios. This process continues until the human is satisfied with the agent's understanding of the task and its safety constraints, at which point the agent can be deployed in the real environment. If all goes to plan, this agent should avoid unsafe states in the real environment without having needed to visit those states in the first place. For further details, see Reddy et al. (2019).

**ReQueST in this work**  ReQueST is flexible in i) the type of feedback used to train the reward model, ii) the proxies for value of information used to elicit informative feedback, and iii) the algorithm used to train the agent in the learned simulation. In this work, for i) we use reward sketching (Cabi et al., 2019), one of the highest-bandwidth feedback mechanisms currently available. For ii), we use maximisation and minimisation of reward as predicted by the current reward model. Finally, for iii) we use model predictive control – appealing for its simplicity, requiring no addi-

tional training once the dynamics model and reward model are complete. See Fig. 2 for details, and Appendix A for pseudocode.

## 3.2 PROBLEM SETTING

Our 3D environment consists of an arena with two apples out in the open, and a third apple behind a gate that must be opened by stepping on a button (shown in Fig. 3). Agents receive $96 \times 72 \times 3$ RGB pixel observations from a first-person perspective, and take actions in a two-dimensional continuous action space allowing movement forward and backward and turns left and right. Agent spawn position, positions of the two apples out in the open, and wall and floor colour are all randomised.

**Task**  The task is to eat the apples by moving close to them, ideally eating all 3 apples. The episode ends when either the agent eats the apple behind the gate or 900 steps have elapsed. To test Re-QueST's ability to avoid unsafe states, we use two variations on this setup.

**Cliff edge environment**  In the first environment variant, we remove the walls from the arena, so that it is possible for the agent to fall off the side. Such a fall is considered unsafe, and immediately ends the episode.

**Dangerous blocks environment**  The second environment variant models a more subtle case: where some part of the agent's internal mechanisms is exposed in environment, and we would like to use safe exploration to discourage the agent from tampering (Everitt et al., 2021) with those mechanisms. We use a scenario based on the REALab framework (Kumar et al., 2020), where instead of training rewards being communicated to the agent directly, rewards are communicated through a pair of blocks, where the reward given to the agent is based on the distance between these blocks. If the agent happens to bump into one of these blocks, changing the distance between them, this will affect the reward the agent receives, likely interfering with the agent's ability to learn the task. Unsafe behaviour in this environment corresponds to making contact with the blocks.

**Sized subvariants**  We further use three subvariants, *small*, *medium* and *large*, of each of these two main variants. In each subvariant, the apples are the same distance from the agent spawn position, but the arena size is different, varying the difficulty. The larger the arena, the easier it is to avoid unsafe states: the edges are further away in the cliff edge variant, and the blocks are further away in the dangerous blocks variant. See Appendix B for details.

## 3.3 DYNAMICS MODEL

**Architecture**  Our dynamics model predicts future pixel observations conditioned on action sequences and past pixel observations using latent states as predicted by an LSTM. See Fig. 4 for an example rollout. Our model is similar to that used in Reddy et al. (2019), except that we use a larger encoder network, a deconvolutional decoder network, and train using a simple mean-squared-error loss between ground-truth pixel observations and predicted pixel observations. Crucially, we base this loss on predictions multiple steps into the future for each ground-truth step (Gregor et al.,

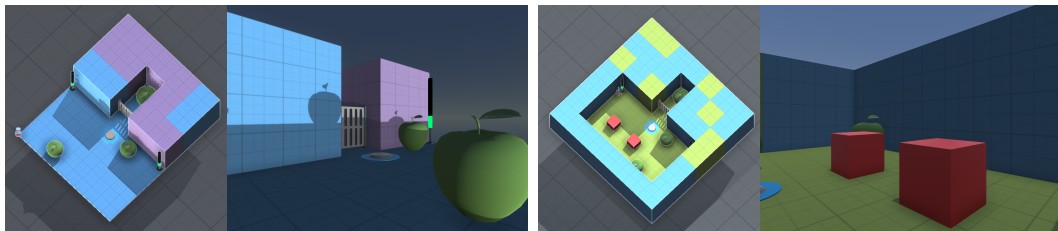

Figure 3: 3D environments used for our experiments. Environments consist of green apples, red blocks, and a gate opened by a switch. Wall and floor colours are randomised. Also shown are the agent's body in the top corner of the environment, and two timer columns, indicating how much time is left in the episode by how much of the column is green. **Left**: cliff edge environment. The agent must avoid falling off the edge of the environment. **Right**: dangerous blocks environment. The agent should avoid interfering with the red blocks, the distance between which encodes the reward sent to the agent.

2019), enabling the dynamics model to remain coherent even over long rollouts. See Appendix D for additional details.

**Data collection** We collect safe exploratory trajectories used to train the dynamics model using a crowd computing platform. All contractors provided informed consent prior to starting the task, and were paid a fixed hourly rate for their time. Note that we instruct contractors to *explore* the environment rather than simply eat apples, in order that the dynamics model fully covers the state space. See Appendix H for contractor instructions, and Table 1 for data requirements.

| Step 0 | Step 30 | Step 70 | Step 90 | Step 100 | Step 130 | Step 180 |

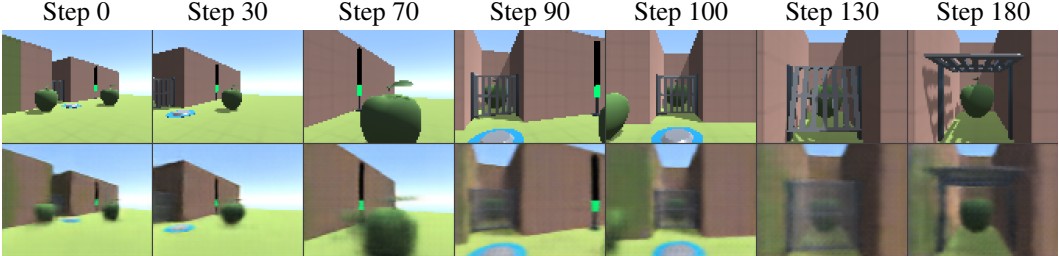

Figure 4: Example rollout from dynamics model using test set initial state and action sequence. We require high-quality rollouts for humans to be able to give meaningful feedback. **Top row**: ground-truth frames. **Bottom row**: frames predicted by dynamics model, conditioned on ground-truth frame at step 0 and (for second column onwards) action sequence. Note that the rollout remains coherent over many timesteps, even modeling the opening of the gate when the agent steps on the button.

### 3.4 REWARD MODEL

**Reward sketches** Our reward models are trained using reward sketches (Cabi et al., 2019): a curve for each episode drawn by a human using the mouse cursor, representing what they think the dense reward between -1 and 1 should be at each moment in time. We instruct contractors to sketch rewards based on the distance to the nearest visible apple – positive increasing rewards when moving closer, positive decreasing rewards when moving away, and negative rewards when close to an unsafe state (the edge of the environment or a red block). See Appendix J and our website for examples.

**Data collection** We collect reward sketches using a crowd compute platform, asking contractors to sketch 50-frame segments of each episode at a time. All contractors provided informed consent prior to starting the task, and were paid a fixed hourly rate for their time. See Appendix H for contractor instructions, and Table 1 for data requirements.

**Architecture and training** Our reward model is a feed-forward network that takes as input individual $96 \times 72 \times 3$ RGB pixel observations predicted by the dynamics model. The network is trained to regress to the sketch values, and the output of the network is then used directly by the agent. See Appendix E for further details on architecture and training.

**Programmatic reward bonus** Because our sketches are based on the distance to the nearest apple, the optimal policy would simply walk up to an apple and stay there. To overcome this, we augment predicted rewards with a programmatically-specified bonus that encourages the agent to actually eat each apple. This bonus gives a reward of +100 when the predicted reward becomes $\geq 0.7$ and then $\leq 15$ steps later drops to $\leq 0.1$. This corresponds to being close to an apple then the apple suddenly disappearing because it has been eaten. Note that the reward model is nonetheless strongly load-bearing, predicting distance to the nearest apple – a function that would be difficult to compute procedurally from pixels. (This bonus is not applied for the baselines, which instead use ground-truth sparse rewards, as discussed below.)

**Trajectory optimisation** We use two iterations of human feedback. First, we train an initial reward model based on sketches from 400 trajectories generated by the dynamics model conditioned on random action sequences. Second, we use the reward model from the first iteration to synthesise a set of 100 more informative trajectories that the reward model believes are high or low reward (50 sketches each). We do this by computing the gradient of the predicted reward through both the reward model and the dynamics model to the actions used to condition dynamics model predictions, then optimise for maximum or minimum reward using gradient descent.

| Model | Data | People | Per person | Total time |
|-------|------|--------|-----------|-----------|
| **Dynamics model** | 15k trajectories $\approx$ 10M frames | 20 contractors | 8 hours | 160 person-hours |
| **Reward model** (First iteration) | 400 reward sketches $\approx$ 20k reward values | 1 contractor | 8 hours | 8 person-hours |
| **Reward model** (Second iteration) | 100 reward sketches $\approx$ 5k reward values | 1 contractor | 2 hours | 2 person-hours |

Table 1: Data requirements. A large amount of data is required to train the dynamics model, but much less for the reward model. Each trajectory contains up to 900 steps – about 15 seconds. Reward sketches are done on video clips 50 steps long, each sketch taking about 90 seconds.

## 3.5 AGENT

We deploy the agent using the learned dynamics and reward model in the real environment using a simple model-based control algorithm, model predictive control (Garcia et al., 1989). On each planning iteration, we sample 128 random trajectories each 100 steps long from the dynamics model. We pick the trajectory with the highest predicted return using a discount of 0.99, take the first 50 actions from that trajectory in the real environment, then replan. See Appendix C for details.

**Baselines** We compare to three baseline agents. The first is alternate ReQueST agent trained on sparse feedback, matching Reddy et al. (2019): +1 when the agent eats an apple, -1 when near an unsafe state, and 0 otherwise. The second is a model-free agent, R2D2 (Kapturowski et al., 2018), which we train using the ground-truth rewards of +1 for each apple eaten (and in the safe exploration environment, a penalty of -1 when falling off the edge). The third is an agent that takes correlated random actions. See Appendix I for details.

## 4 RESULTS

We evaluate ReQueST by running MPC for 100 episodes in each environment variant. For the model-free baseline, we train just up to convergence, and then examine metrics from the most recent 100 evaluation episodes. For the random-actions baseline, we simply run for 100 episodes. In all cases, we run 10 different seeds and report the median, with error bars showing the 25th and 75th percentiles. Videos are available at https://sites.google.com/view/safe-deep-rl-from-feedback.

### 4.1 CLIFF EDGE ENVIRONMENT

In the cliff edge environment, we are able to use ReQueST to train a competent agent with between 3 and 20 times fewer instances of unsafe behaviour than by training with a traditional model-free RL algorithm (see Fig. 5). Moreover, the bottleneck is actually the safety of the human-demonstrated trajectories used to train the dynamics model. For these experiments, we didn't try very hard to minimize human safety violations. Since our training procedure makes very few errors at deployment time, reducing or eliminating human error through e.g. incentives for contractors to be more careful would allow us to train an agent with very close to *zero* instances of unsafe behaviour.

In terms of performance, the ReQueST agent eats 2 out of the 3 apples on average. This is significantly better than the random baseline, which eats on average only 1 out of the 3 apples, and falls off the edge in about half of the 100 evaluation episodes. The model-free baseline (trained using ground-truth sparse rewards) eats all 3 apples, but at the cost of safety: even in the best case, it must fall off the edge over 900 times before it learns *not* to.

### 4.2 DANGEROUS BLOCKS ENVIRONMENT

Results for our dangerous blocks environments are shown in Figure 6. Here the model-free baseline is barely able to perform the task at all, instead simply tampering with the blocks (as shown by the large number of safety violations). This is because the agent moves blocks so that the distance

between them is as large as possible, causing the rewards the agent receives to be set permanently high until the end of the episode (when the blocks are reset). Furthermore, since the reward channel is saturated, rewards from eating apples can no longer be observed. ReQueST is the *only* agent capable of performing the task in these environments – with a greater number of safety violations than the safe exploration environments only because of the high difficulty of avoiding the blocks in the smaller environment sizes.

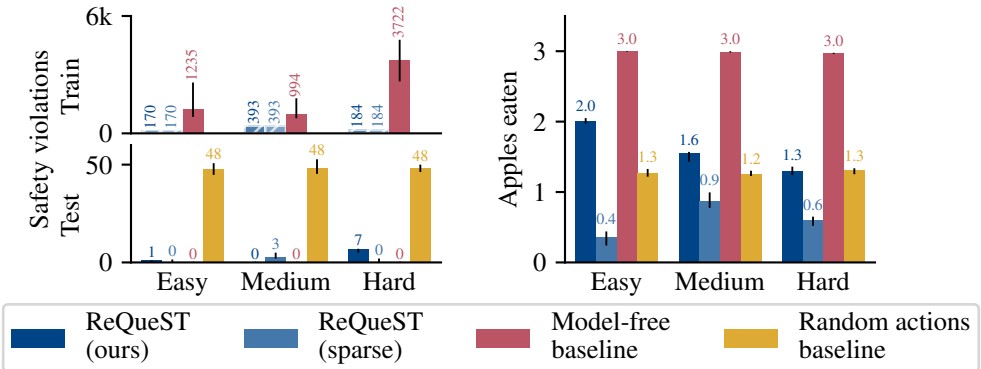

Figure 5: Cliff edge environment results. **Left**: safety, measured by the number of times the agent fell off the edge of the environment. Note that ReQueST train-time safety violations consists entirely of mistakes by human contractors while providing demonstration trajectories. **Right**: performance, measured by the average number of apples eaten per episode. Each difficulty represents a different arena size, with larger arenas being easier. Runs are repeated over 10 seeds, with bar heights/labels showing medians and error bars showing 25th/75th percentiles.

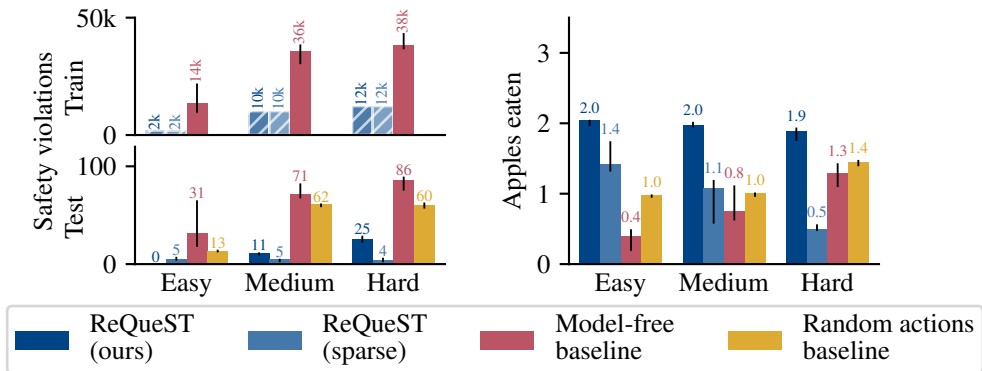

Figure 6: Dangerous blocks environment results. **Left**: safety, measured by the total number of times the agent touched the blocks. Note that ReQueST train-time safety violations consist entirely of mistakes by human contractors while providing demonstration trajectories. **Right**: performance, measured by the average number of apples eaten per episode.

## 4.3 SCALING

How much data is required to achieve good performance, for both the dynamics model and the reward model? We answer this question by training a number of models with varying sizes of dataset. See Fig. 7 for these results. The main takeaway is that with less data, performance decreases gradually, rather than catastrophically. Our dynamics model appears to follow previously-established trends (Henighan et al., 2020) in which performance varies smoothly with the dataset size according to a power law. For the reward model, however, agent performance appears to plateau after roughly 250 sketches.

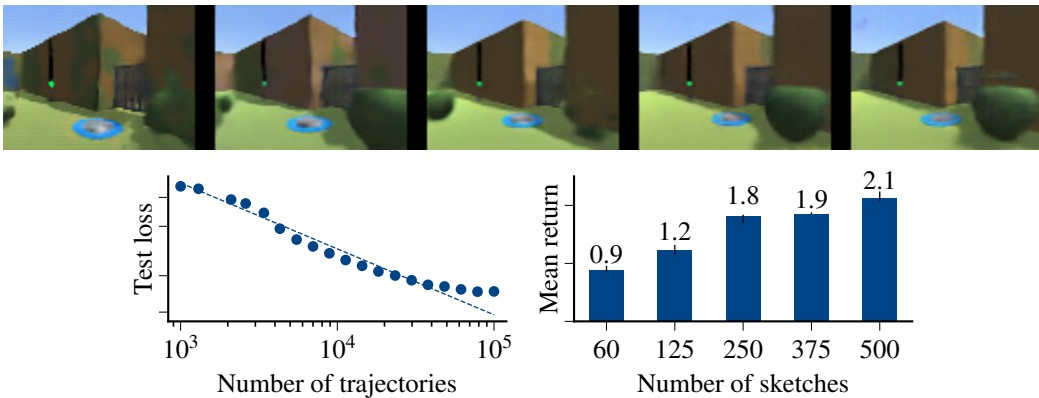

Figure 7: Scaling results for dynamics model and reward model. Top: representative predictions from dynamics models trained on 1k, 3k, 9k, 30k, and 100k trajectories. Left: final smoothed test losses from training dynamics models on 19 dataset sizes. See Appendix D for details. Right: performance of ReQueST agent using reward models trained on different fractions of full sketches dataset (comprised of sketches on both random and optimised trajectories). Bar height shows median apples eaten in 100 evaluation episodes over 10 seeds; error bars show 25th and 75th percentiles.

## 5 DISCUSSION

From these results, should we consider ReQueST a realistic, scalable approach to safe exploration?

### 5.1 ASSUMPTIONS

ReQueST relies on the availability of a sufficient quantity of safe exploratory trajectories. Though we have shown the required quantity to be practical (see Table 1) for a 3D environment of moderate complexity, it is difficult to say how well our numbers would generalise to a real-world task (e.g. dishwasher loading). Also, not every task will be amenable to the parallelisation among many humans that made the data requirements viable in our case. For example, the task may be difficult enough that only a small number of expert humans are capable of safely navigating the state space, such as helicopter flight. This is especially true when dynamics are non-uniform, such as helicopter aerobatics. Here, merely exploratory trajectories would be insufficient; demonstrations of the aerobatic manoeuvres themselves would be needed to learn the unusual parts of the state space. For some tasks, however, this issue may be mitigated by unsupervised pretraining (Stiennon et al., 2020).

ReQueST also relies to some extent on the existence of near-unsafe states. Not all tasks contain such states; for example, if a robot is gripping a porcelain cup high above a stone floor, the switch between safe and unsafe is near-instantaneous. In some cases it may be possible to train the dynamics model on similar but low-stakes trajectories (e.g. dropping plastic cups) hoping that the dynamics model will generalise to high-stakes scenarios. In other cases, it is possible that a ReQueST agent would stick close enough to the positive paths (carrying the cup while keeping the grippers closed) that being pushed away from the negative paths (dropping the cup) is less important. However, this will not always work – in our dangerous blocks environments, for example, the unsafe blocks are *on the way* to the apples, so repulsion from unsafe blocks *is* necessary.

Finally, ReQueST assumes that unsafe states (or a superset of states surrounding unsafe states) can be recognised by humans. This assumption is likely reasonable for tasks of current practical interest. However, it may be too strong an assumption looking to the future, particular for tampering: future AI systems may find ways to tamper that are pernicious exactly because they can *not* be recognised by humans. To guard against these kinds of issues, we will need further progress on scalable oversight techniques, e.g. Christiano et al. (2018); Irving et al. (2018); Leike et al. (2018).

## 5.2 ALTERNATIVES

**Imitation learning** First, note that in imitation learning, performance is limited by the quality of the demonstrations (Mandlekar et al., 2021), while the RL-based nature of our approach should enable superhuman performance to be reached. Second, note that demonstration quality is less of an issue for our approach in the first place. We do not require optimal demonstrations, or even necessarily demonstrations of the task at all: to train the dynamics model we require only a set of trajectories that provide good coverage of the state space. (However, this does require *more* data than is needed for imitation learning, which does not not require knowledge of the whole state space.)

**Offline reinforcement learning** Since we assume the environment does not itself provide rewards, for offline reinforcement learning we would, as in this work, need to use rewards from a reward model trained using e.g. reward sketches on a subset of the human demonstration trajectories (Cabi et al., 2019; Konyushkova et al., 2020; Zolna et al., 2020). However, note that in this case, assuming no simulator is available, the agent could not be evaluated without running it in the real environment, and thus there might be no way to estimate the safety of such an agent without risking an accident in the real world.

**Constrained reinforcement learning** As noted in Section 2, a common approach to safe exploration is to provide a procedural constraint function which the agent should violate as little as possible during training. Replacing this procedural function with a learned constraint function would make for an interesting comparison to the present work. However, note that similarly to offline reinforcement learning, such an agent could not be safely evaluated without access to a simulator.

## 6 CONCLUSION

In this work we have shown that ReQueST can be used to train an agent in a 3D environment with an order-of-magnitude reduction in instances of unsafe behaviour than typically required with reinforcement learning. If unsafe behaviour can be avoided by humans (or trajectories are collected by watching humans carry out tasks they are already doing, such that no *additional* safety violations are incurred) our method can come close to achieving *zero* instances of unsafe behaviour. Moreover, this is possible without a procedural specification of safe behaviour, and with minimal assumptions other than unsafe or near-unsafe states being recognisable by humans.

Our results are exciting for two reasons. First, they demonstrate that ReQueST is plausibly a general-purpose solution to (one version of) the safe exploration problem. Arguably, this is the version we're most interested in long-term: where safe behaviour is learned from humans, rather than given by an easy-to-misspecify procedural function. Some tasks will require higher-fidelity dynamics models than those used in this work, but given recent advancements in generative image modelling (Karras et al., 2018; 2019; Mildenhall et al., 2020), we are optimistic for future progress in this area. Second, our results hint at a future where our ability to verify agent behaviour before deployment is not bottlenecked by the availability of a simulator – the simulator can be *learned* from data.

In terms of future work, one particularly important question is: what challenges arise when trying to use ReQueST in a real-world environment, such as a robotics task? We look forward to future work addressing this line of research.

## 7 ETHICS STATEMENT

**Contractor welfare** Our method relies on a large quantity of data from human contractors, requiring them to perform a task that is extremely monotonous for several hours per person. Additionally, during one round of feedback our contractors reported mild motion sickness due to latency on our interface. Those wishing to use our method in practice should ensure that contractors are well-compensated; that they feel free to take as many breaks as they need; and ideally that no individual contractor is required to perform the same task for too long (relying on a large number of contractors rather than long sessions in order to gather sufficient data).

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

# Appendices

## A  PSEUDOCODE

Denoting:

- Observation $o$ from observation space $\Omega$. Observation predicted by dynamics model $\hat{o}$.

- Action $a$ from action space $\mathcal{A}$. Action sequence $A$. Optimised action sequence $A^*$.

- Trajectory $\tau$. Sequence of observations predicted by dynamics model $\hat{\tau}$. Sequence of predicted observations sent to user for reward sketching $\hat{\tau}_{query}$.

- Reward sketch value $r$. Distribution of reward sketch values assigned by human $p_{user}$.

---

**Algorithm 1:** Dynamics and reward model training

---

**Data:** Safe trajectories $\mathcal{M}_{train}$, $\mathcal{M}_{test}$; Reward sketches $\mathcal{D} \leftarrow \emptyset$
**Parameters:** Dynamics model parameters $\phi$, Reward model parameters $\theta$
**Models:** Dynamics model $D_\phi : \Omega \times \mathcal{A} \to \Omega$, Reward model $R_\theta : \Omega \to \mathbb{R}$
**Hyperparameters:** Number of reward sketches to obtain on each iteration $N_{sketches}$
// Train dynamics model
1   $\phi \leftarrow \arg\min_\phi \text{MSE}[\tau \sim \mathcal{M}_{train}, \hat{\tau} \sim D_\phi(\cdot|o_0, a_0, a_1 \dots \sim \tau)]$
// Train reward model (convergence judged by human)
2   **while** $\theta$ *not converged*
3     **repeat** $N_{sketches}$ **times**
4       $o_0 \sim \mathcal{M}_{test}$
5       **if** $\theta$ *not initialised*
6         $\hat{\tau}_{query} \leftarrow \hat{o}_0, \hat{o}_1, \dots \sim D_\phi(\cdot|o_0, A \sim \text{RANDOM})$
7       **else**
8         $A^* \leftarrow \arg\max_A \sum_{\hat{o}_t \sim D_\phi(\cdot|o_0, A)} R_\theta(\hat{o}_t)$
9         $\hat{\tau}_{query} \leftarrow \hat{o}_0, \hat{o}_1, \dots \sim D_\phi(\cdot|o_0, A^*)$
10       **for** $\hat{o} \in \hat{\tau}_{query}$
11         $r \sim p_{user}(r|\hat{o}, \hat{\tau}_{query})$
12         $\mathcal{D} \leftarrow \mathcal{D} \cup (\hat{o}, r)$
13     $\theta \leftarrow \arg\min_\theta \sum_{(\hat{o},r) \in \mathcal{D}} \text{MSE}[r, R_\theta(\hat{o})]$

---

- Initial real environment state $s_0$ sampled from start-state distribution $\rho_0$. Subsequent states $s_t$ computed by state transition function $f(s_{t-1}, a)$.

- Observation conditioned on real environment state $\mathcal{O}(s)$.

- Random action sequence for $n$th rollout $A^n$. Action at index $k$ within each sequence $A_k^n$.

---

**Algorithm 2:** Model predictive control

---

**Models:** Dynamics model $D_\phi : \Omega \times \mathcal{A} \to \Omega$, Reward model $R_\theta : \Omega \to \mathbb{R}$
**Hyperparameters:** Number of sample rollouts per replan $N_{samples}$, Steps per sample
                      rollout $N_{plan}$, Discount $\gamma$, Number of steps after which to
                      replan $N_{replan} \leq N_{plan}$
1   Initialise $t \leftarrow 0$, $s_0 \sim \rho_0$
2   **while** $s_t$ *not terminal*
3     $o_t \leftarrow \mathcal{O}(s_t)$
4     $A^n \sim \text{RANDOM} \; \forall \, n \in [1, N_{samples}]$
5     $A^* \leftarrow \arg\max_{A^n} \sum_{k=1}^{N_{plan}} \gamma^{k-1} R_\theta(\hat{o} \sim D_\phi[\cdot|o_t, A_k^n])$
6     **for** $a \in a_t, \dots, a_{t+N_{replan}} \subseteq A^*$
7       $s_{t+1} \leftarrow f(s_t, a)$   // Take action in environment
8       $t \leftarrow t + 1$

---

See Appendix C for more details on our model predictive control implementation.

# B    ENVIRONMENT DETAILS

Our six total environment variants are illustrated in Fig. 8. Our environments are implemented using Unity, with randomisation of wall and floor colours to improve generalisation.

In the rest of this appendix, we discuss spawn positions.

## B.1    APPLES

The two apples that are out in the open always spawn in the same area, near the gate, regardless of the size of the arena. This ensures that the margin for error increases as the size of the arena increases: the larger the arena, the further away the unsafe parts of the arena (the edges of the arena/the red blocks) are from the important parts of the task, the apples. Spawn positions are selected randomly from a discrete set of eight possible locations.

## B.2    BLOCKS

In the dangerous blocks environment, the blocks spawn in a random position around a semicircle centred on the button by the gate. The radius of the semicircle increases with the size of the arena, such that the larger the arena, the further away that blocks spawn.

## B.3    AGENT

In the **safe cliff edge** environments (top row of Fig. 8), the agent can spawn in one of the four corners of the smallest arena size: by either of the two timer columns, or in either of the two corners surrounded by open space. These spawn locations remain unchanged in the larger sizes of environment, so that when the arena is larger, the agent starts further from the edge. The agent always spawns looking towards the centre of the arena, such that the first frame of the episode is more likely to show both apples. This is important for helping the dynamics model to establish the positions of the apples.

In the **dangerous blocks** environments, the agent can only spawn by either of the two timer columns – again, looking out towards the centre of the arena. Otherwise, the red blocks may not be visible in the first frame of the episode, which causes the dynamics model to 'hallucinate' blocks in arbitrary locations.

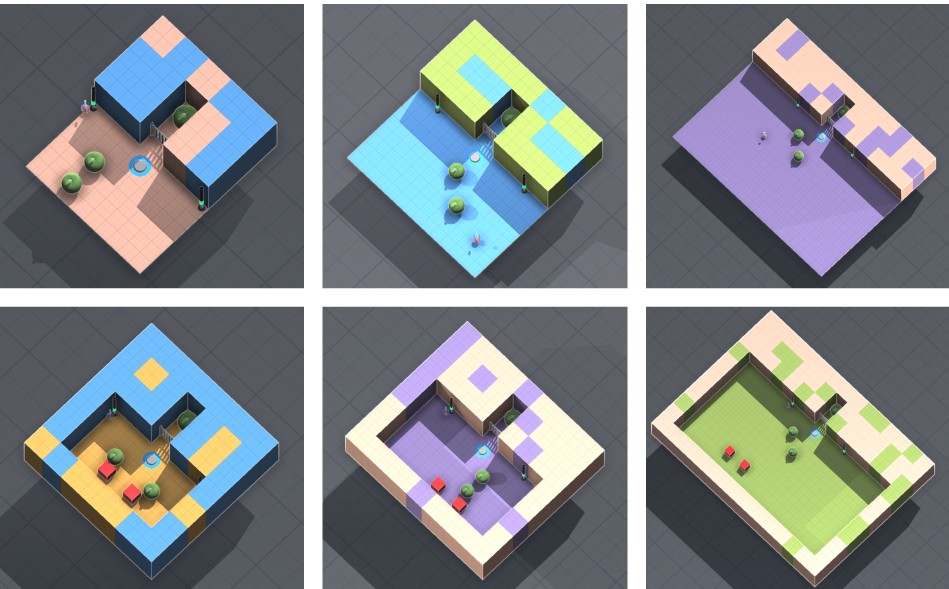

Figure 8: All six of our environments variants, showing the three possible arena sizes. First column: small. Second column: medium. Third column: large.

## C  MODEL PREDICTIVE CONTROL

We implement model predictive control with sampling-based optimisation. This is because we augment predicted rewards using a non-differentiable procedural reward bonus (see Section 3.4). This functions as follows.

First, we generate $N_{samples} = 128$ random action sequences $A$, each $N_{plan} = 100$ steps long. Each action sequence starts with a random number (between 0 and $N_{plan}$) of 'turn left' or 'turn right' actions, then all remaining actions are 'move forward'. (Note that we try to match this for the random-actions baseline; see Appendix I.)

$$\forall\, n \in (1, ..., N_{samples})$$
$$a_1, ..., a_{N_{plan}} \sim \text{RANDOM}$$
$$A^n \leftarrow (a_1, ..., a_{N_{plan}})$$

For each action sequence, we generate an imagined rollout starting from the current state $s_t$ in the real environment. This involves first initialising the latent state of the dynamics model $l_0$ by feeding the history of observations from the real environment in the current episode $o_1, ..., o_t$ through our encoder network $E$.

$$l_0 \leftarrow E(o_1, ..., o_t)$$

We then use the dynamics model $D$ to predict future states $l_1, l_2, ...$ conditioned on each action $a$ from the sequence. We use the decoder network $F$ to transform each latent state to a predicted pixel observation $\hat{o}$, then use the reward model $R$ to predict a reward $\hat{r}$ for each predicted observation. This yields a predicted return $\hat{R}^n$ (discount $\gamma = 0.99$) for each action sequence $A^n$.

$$l_1 \leftarrow M(l_0|a_1) \qquad \hat{o}_{t+1} \leftarrow F(l_1) \qquad \hat{r}_{t+1} \leftarrow R(\hat{o}_{t+1})$$
$$...$$
$$l_{100} \leftarrow M(l_{99}|a_{100}) \qquad \hat{o}_{t+100} \leftarrow F(l_{100}) \qquad \hat{r}_{t+100} \leftarrow R(\hat{o}_{t+100})$$

$$\hat{R}^n \leftarrow \sum_{k=1}^{100} \gamma^{k-1}\, \hat{r}_{t+k}$$

We then select the action sequence with the best predicted return, and take the first $N_{replan}$ steps in the environment, and repeat.

$$N \leftarrow \arg\max(\hat{R}^n)$$
$$A^* \leftarrow A^N$$
$$\texttt{for a in } A^*[:N_{replan}]\texttt{: env.step(a)}$$

We selected the hyperparameters $N_{samples}$, $N_{plan}$ and $N_{replan}$ using grid search. Rollouts are run in parallel using 50 CPU workers, with each worker running on a separate cluster node. Using this setup, each episode of 900 steps takes about 15 minutes.

## D  DYNAMICS MODEL

Our dynamics model consists of three components: i) a recurrent encoder network that transforms a history of pixel observations into an initial latent state, ii) a recurrent next-state network that computes future latent states conditioned on actions, and iii) a feed-forward decoder network that transforms a latent state back into a predicted pixel observation. The model consists of 50M parameters total.

### D.1  ENCODER NETWORK

The encoder network computes a latent state corresponding to the current state of the real environment, taking as input the history of observations from the current episode in the real environment. To do this, we first process each observation using a residual network (He et al., 2016) as constructed by the following pseudocode:

```
x = observation
for num_channels in [16, 32, 32]:
  x = conv_2d(x, kernel_shape=3, stride=1, channels=num_channels)
  x = max_pool(x, window_shape=3, stride=2)
  for _ in range(2):
    block_input = x
    x = relu(x)
    x = conv_2d(x, kernel_shape=3, stride=1, channels=num_channels)
    x = relu(x)
    x = conv_2d(x, kernel_shape=3, stride=1, channels=num_channels)
    x += block_input
x = relu(x)
```

For $N$ observations, this yields $N$ outputs. However, each of these outputs is a function of only a single observation. To produce a latent state which is a function of the *history* of observations, we additionally apply a single-layer LSTM (Hochreiter & Schmidhuber, 1997) with 1024 hidden units over the sequence of these outputs, and take the final output from the LSTM to produce the latent state. (Note that this is a *separate* LSTM than the one used for the next-state model.)

The encoder residual network consists of 100k parameters, and the encoder LSTM consists of 18M parameters.

### D.2  NEXT-STATE MODEL

The next-state model computes future latent states using the initial latent state and a sequence of actions. This is also implemented using a single-layer LSTM with 1024 units. The LSTM consists of 4M parameters.

### D.3  DECODER NETWORK

The decoder takes a single latent state and transforms it back into a pixel observation. It is a feed-forward deconvolutional network using FiLM conditioning (Dumoulin et al., 2018) as constructed by the following pseudocode:

```
first_output_shape = [3, 4, 1024]
output_shapes = [(9, 12), (18, 24), (36, 48), (72, 96), (72, 96)]
output_channels = [512, 256, 128, 64, 3]
strides = [3, 2, 2, 2, 1]
kernel_shapes = [4, 4, 4, 4, 3]

x = latent
x = linear(x, output_size=np.prod(first_output_shape))
x = reshape(x, first_output_shape)

for layer_num in range(5):
```

```
  prev_layer_num_channels = x.shape[-1]
  offset = linear(
    latent,
    output_size=prev_layer_num_channels,
  )
  # Scale should have a mean of 1.
  scale = 1 + linear(
    latent,
    output_size=prev_layer_num_channels,
  )
  x = x * scale + offset
  x = leaky_relu(x)
  x = conv_2d_transpose(
    x,
    kernel_shape=kernel_shapes[layer_num],
    stride=strides[layer_num],
    output_shape=output_shapes[layer_num],
    output_channels=output_channels[layer_num],
  )
x = sigmoid(x)
```

In total, the decoder network consists of 28M parameters.

## D.4 Loss

We compute the loss based on the MSE between the predicted and actual pixel observations. Additionally, we use latent overshooting (Gregor et al., 2019) to improve long-term coherence, and subsampling to save on GPU memory. The overall procedure is as follows.

```
trajectory_num_steps = len(observations)
steps_to_start_predicting_from =
  random.choice(
    trajectory_num_steps,
    size=num_prediction_sequences,
  )

loss = 0
for starting_step in steps_to_start_predicting_from:
  latent = encoder(observations[:starting_step])
  predicted_latents = []
  for _ in range(predictions_length):
    latent = next_state_model(latent)
    predicted_latents.append(latent)
  steps_to_decode = random.choice(
    predictions_length,
    size=num_prediction_steps_sampled,
  )
  for step_to_decode in steps_to_decode:
    latent = predicted_latents[step_to_decode]
    predicted_observation = decoder(latent)
    actual_observation = observations[starting_step + step_to_decode]
    loss += (predicted_observation - actual_observation) ** 2
```

We use num_prediction_sequences=6, predictions_length=150, and num_prediction_steps_sampled=10.

## D.5 TRAINING

We train using the Adam optimizer (Kingma & Ba, 2014) with a learning rate of $2 \times 10^{-4}$ and a batch size of 48 trajectories (about 43k individual steps).

We train using 8 GPUs (with batch parallelism) for 1M steps – about 2 weeks. We train a total of 6 dynamics models – for each of the 6 environment variants (see Section 3.2). For all 6 models, the final train loss is about 15, and the final test loss is about 30.

## D.6 SCALING

Recent work has found performance of large models to vary surprisingly predictably as a function of, for example, dataset size (Henighan et al., 2020). We wanted to check whether this was true for our dynamics model too, so we trained a number of dynamics models on varying dataset sizes, as shown in Fig. 7.

This figure was produced as follows. First, we generated a synthetic dataset of 100,000 trajectories in the real environment comprised of random action sequences. We then trained 20 different dynamics models on logarithmically-spaced fractions of this dataset, all models using the same hyperparameters. We trained until all models' test losses had definitely plateaued – about 1.2M steps, 17 days. The model trained on 1,600 trajectories did not train properly so we ignored it. We then smoothed the test loss curve by taking an exponential moving average (smoothing factor $\alpha = 0.001$), and took the minimum of this smoothed curve to be the loss value reported for each model. See Fig. 9 for training curves.

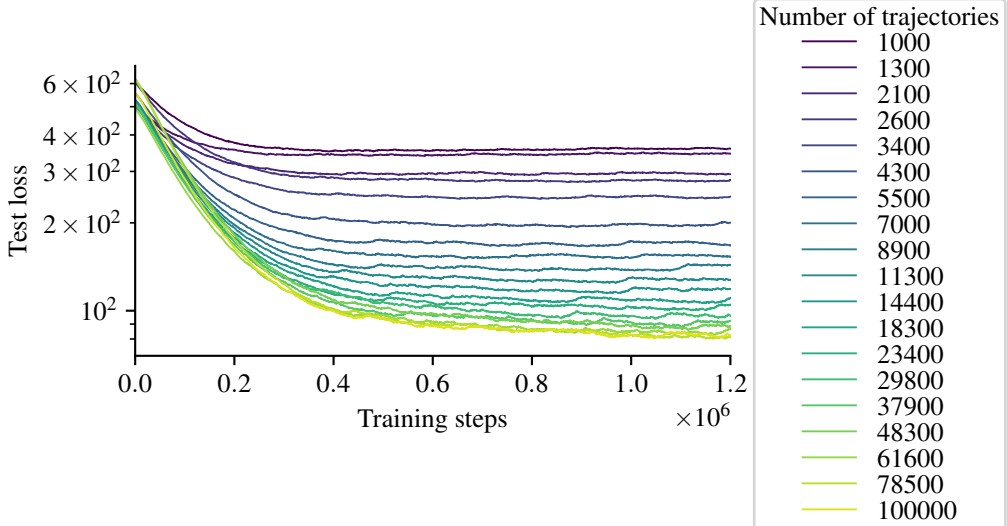

Figure 9: Dynamics model training curves.

## E    REWARD MODEL

Our reward model is a 2.2M parameter 11-layer convolutional network with residual connections (He et al., 2016) that takes an RGB $96 \times 72$ input and produces a scalar reward prediction, constructed as follows:

```
x = observation
for num_channels in [16, 32, 32]:
    x = conv_2d(x, kernel_shape=3, stride=1, channels=num_channels)
    x = dropout(x, rate=0.1)
    x = max_pool(x, window_shape=3, stride=2)
    for _ in range(2):
      block_input = x
      x = relu(x)
      x = conv_2d(x, kernel_shape=3, stride=1, channels=num_channels)
      x = dropout(x, rate=0.1)
      x = relu(x)
      x = conv_2d(x, kernel_shape=3, stride=1, channels=num_channels)
      x += block_input
      x = dropout(x, rate=0.1)
x = relu(x)
x = flatten(x)
x = linear(x, num_units=128)
x = dropout(x, rate=0.1)
x = relu(x)
x = linear(x, num_units=128)
x = dropout(x, rate=0.1)
x = linear(x, num_units=1)
predicted_reward = x
```

We train using a mean-squared error loss, using the AdamW optimiser with hyperparameters shown in Table 2. We train using a batch size of 64 (where each data point consists of one observation and the corresponding sketched reward value), for a total of 20,000 batches in all cases (which was consistently sufficient for the validation loss to plateau). Training using a single GPU, this takes about 1 hour.

Batches must be sampled carefully due to dataset imbalances. For example, when collecting reward sketches from random trajectories, only a small number of those trajectories will actually show the apple being eaten. To account for the imbalance, we adopt the following procedure. First, we split all sketches into individual timesteps, and collect the timesteps in a pool. Second, we split the pool into groups of timesteps, where each group represents one type of situation – in our case, i) being far away from both apples and hazards, ii) being close to a hazard, and iii) being close an apple – as determined by the sketched reward value for that timestep. Finally, when sampling a batch, we sample a roughly equal number of timesteps from each group (placing slightly more weight on the 'close to an apple' group, which is the most important), ensuring that each batch contains data for all situations of interest. See Table 3 for numbers used for this procedure.

| Hyperparameter | Value |
|---|---|
| Weight decay | $2 \times 10^{-4}$ |
| Learning rate | $8 \times 10^{-4}$ |
| $\beta_1$ | 0.7 |
| $\beta_2$ | 0.9 |

Table 2: AdamW hyperparameters.

| Situation | Value range | Batch proportion |
|---|---|---|
| Close to apple | 0.7 to 1.0 | 0.4 |
| Close to hazard | -1.0 to -0.5 | 0.3 |
| Close to neither | -0.5 to 0.7 | 0.3 |

Table 3: Batch sampling values.

# F  EVALUATING DATA AND MODEL QUALITY

By learning in a neural simulation, ReQueST aims to produce an agent that is safe to deploy directly in the real world. But how can we be sure of this – how do we know when the agent has been trained enough that it really is both competent and safe?

In principle, the model-based nature of ReQueST allows this determination to be made by simply running agent rollouts in the neural simulation. In practice, this requires that the dynamics model both a) remains coherent over the entire length of a simulate episode, and b) matches the real environment closely; criteria that the dynamics models we trained in this work do not meet. Further progress in generative video modelling will be needed for this ambition to be realised.

In the meantime, we can get a feel for how the agent is likely to behave by assessing the performance of the individual models that the agent relies on. In this appendix, we detail the methodology we used to do to this, for both the models themselves and the data used to train them.

## F.1  EXPLORATORY TRAJECTORIES

A model is only as good as the data it's trained on. To ensure the data used to train the dynamics model was of sufficient quality, we found the following three checks helpful.

**Qualitative inspection**  Simply viewing videos of the trajectories, making it easy to get through many videos quickly by arranging videos in a grid and watching at high speed, was an important first step in getting a sense for what's going on in the trajectories and what problems might exist in them.

**Action counts**  Given how dull the task is, it's not surprising that the quality of trajectories produced by contractors was somewhat mixed. In particular, some contractors would spend some trajectories mostly idle, or just moving consistently in a straight line. To ensure such trajectories did not make up a significant fraction of the dataset, we found it helpful to plot histograms of a) the number of no-op actions per trajectory; b) the number of times the action changed per trajectory; and c) the number of distinct actions per trajectory.

**State space coverage**  It's important that the trajectories cover all relevant parts of the state space – for example, that there are a good number of trajectories in which the apple is eaten. In our case we were able to do this by examining the distribution of ground-truth rewards provided by the Unity environment. For real-world environments, a good approach here would be to train classifiers for various situations from the trajectories, and examine the distribution of those classifiers' outputs over the dataset.

## F.2  DYNAMICS MODELS

There are two main techniques we used to evaluate dynamics models.

**Test set trajectories**  We do not train the dynamics model on all human exploratory trajectories; we reserve some trajectories in a test set to use for evaluation. This test set can be used to evaluate the dynamics model quantitatively, comparing the test loss to the train loss, and qualitatively, manually inspecting the predicted rollouts produced by the dynamics model given a test set initial state and action sequence.

**Interactive exploration**  It is also possible to explore the simulated environment produced by the dynamics model interactively, starting from an initial state sampled from the test set and then generating actions in real time using keyboard input. This allows for fine-grain investigation of specific details that may not be covered by action sequences in the test set – for example, what happens if one walks off the edge of the cliff edge environment backwards?

The combination of these two techniques worked great, and allowed us to be confident in our assessment of dynamics model quality.

### F.3 REWARD SKETCHES

As with the dynamics models, the quality of the reward models depends on the quality of the reward sketches the models are trained on. To ascertain sketch quality, we used the following techniques.

**Qualitative inspection**  As with trajectories, taking a quick look over the sketches themselves, plotting them in a grid to make it easy to look through them quickly, was an important first step in our process.

**Sketch statistics**  Next, we examined the number of sketches where i) all sketch values are close to zero, ii) any sketch value is greater or iii) less than zero, and iv) any sketch value is greater than 0.7 or v) less than -0.7. We also plotted the total number of individual timesteps (rather than the number of trajectories) matching the previous criteria. Finally, we plotted histograms of i) all sketch values pooled, ii) the return for each sketch, iii) the maximum and iv) minimum value in each sketch, v) the difference between the highest and lowest value in each sketch, and vi) the number of distinct binned values in each sketch.

### F.4 REWARD MODELS

We evaluated reward models using similar techniques to the dynamics model.

**Interactive probing**  As with the dynamics model, we also found it helpful to play around with reward models interactively. We did this by running the dynamics model interactively and overlaying a graph of the predicted rewards. This enabled us to quickly check e.g. how consistently the reward model responded to apples, whether the reward model was responsive to the apple behind the gate when the gate is closed, and so on.

**Test set feedback**: Again, we do not train on all sketches, instead reserving some in order to compute a test loss. Due to dataset imbalance described in Appendix E, the test loss must be computed carefully, as different situations will not be represented evenly in the dataset. For example, when collecting sketches of random trajectories, only a small fraction of trajectories will show an apple being eaten; splitting the dataset naively could lead to all these instances being placed in the train set. We therefore split the dataset by first pooling and grouping the data as described in Appendix E, and then constructing split each individual group into a train set and a test set. This ensures that the test loss is representative of all situations of interest.

In practice, we found reward model test loss to correlate relatively poorly with agent performance. We believe the main reasons for this are:

**Coverage**  Because of the way we generate trajectories to be sketched (in the first stage using random action sequences, and in the second stage by optimising for maximum/minimum predicted reward), the set of states covered by the trajectories is not exhaustive.

**Feedback quality**  Because of the way our reward bonus works (triggering when the predicted reward exceeds 0.7 and then quickly drops), our agent is unfortunately quite sensitive to exact reward values, which can vary from sketch to sketch. This is partly a function of contractors' diligence, but mostly a function of a) the difficulty of judging distances to apples consistently, exacerbated by b) the somewhat low fidelity of the trajectories generated by the dynamics model.

To mitigate both of these problems, we briefly experimented with:

**High-quality test set**  This involved manually demonstrating a set of easy-to-sketch trajectories in the neural simulation (starting far away from an apple and walking straight towards it), then carefully sketching the resulting rewards ourselves. However, this turned out to be sufficiently time-consuming (needing to be done for all six environment variants) that we judged it not to be worth the effort.

We ultimately didn't feel very satisfied with these techniques. Interactive probing provided the strongest signal, but due to its interactive nature was unsuitable for evaluating large numbers of models when doing e.g. hyperparameter sweeps. In the end, we chose reward model hyperparameters on the basis of resulting agent performance in the 'real' environment – a technique not applicable to a real-world scenario. If we'd had more time, taking the time to construct a high-quality test set would probably have been the most principled solution.

## G    UNDERSTANDING AGENT FAILURES

Even after having carefully evaluated the data and models that the agent relies on (Appendix F), the agent still may behave poorly at deployment. This appendix details the methodology we developed to understand agent failures, and the failure modes we observed for our task.

### G.1    DEBUGGING METHODOLOGY

We used two main techniques to understand failures.

**Interactive probing from a failed state**    As described in Appendix F, a key tool is the ability to run the dynamics model interactively, providing actions using keyboard input and watching how a graph of the rewards predicted by the reward model changes depending on the situation. When evaluating the models, we initialise the dynamics model from an arbitrary state sampled from the dataset of exploration trajectories. However, it is also possible to initialise the model from a state from a real-world trajectory. For example, if the deployed agent moves towards an apple but stops before getting close enough to eat it, we can initialise an interactive simulation from that state, and explore to try and understand the problem – is it that the reward model does not give additional rewards for moving closer? Is it that the dynamics model loses coherence when moving towards the apple? And so on.

**Examining counterfactual rollouts**    To recap, model predictive control functions by sampling a number of rollouts from the dynamics model, then picking the rollout with the highest predicted reward, and taking the actions from that rollout in the real environment. If the agent takes poor actions, a second possible debugging strategy is therefore to examine the other rollouts which the agent didn't pick. For example, in one case an apple would be in view but the agent would simply turn around rather than going to the apple. By examining the counterfactual rollouts, we were able to see that the agent did *consider* going to the apple, but chose not to because the reward model did not respond to the apple.

### G.2    FAILURE MODES

Using the tools above, we identified the following failure modes.

**Dynamics model fidelity**    One class of failure was caused by the dynamics model failing to generate crisp predicted observations. This affected not only agent performance but also the quality of human feedback. The main two failures modes we saw were a) blurriness and mirage-like effects, where apples would shimmer in and out of existence as the agent moved around the environment, and b) loss of coherence, where long sequences of predictions would sometimes result in the scene dissolving into a jumble of colours.

Dynamics model quality could likely be improved to some extent by more (or better-quality) data, as shown by the results in Fig. 7. However, the same figure suggests that the improvements from more data plateau around 100k trajectories, and even the model we trained on that number of trajectories still exhibited these issues to some extent – leading us to believe that architectural limitations of our model are probably the bottleneck here.

**Reward model mistakes**    The second class of failure was the reward model not being robust enough. This took several forms. In the most egregious cases, the reward model would simply not respond to an apple being clearly in view – though this was rare. The more common failures were a) the predicted rewards fluctuating as the observations predicted by the dynamics model themselves fluctuated – for example, as apples shimmered partly into and out of existence; b) the shaping of rewards being incorrect – for example, the predicted reward not increasing monotonically when moving towards an apple; and c) the precise value of the predicted rewards being incorrect (at least according to the specification we laid out in our instructions to contractors – see Appendix H), in a way that caused our reward bonus not to trigger.

Overall, we suspect the difficulties here were mainly due to poor fidelity of the dynamics model. There could be some lessons here related to the difficulty of accurately and consistently sketching precise reward values, but problems here could equally be explained by irregularities in the output of the dynamics model, so we hold off on making any firm recommendations here.

**Reward bonus hacking**   The reward bonus itself caused a number of problems. In particular, the bonus was sensitive to reward hacking – the agent managing to trigger the bonus in a manner other than the intended one. For example, the agent would raise the predicted reward above the required threshold by moving close to an apple, but then cause the predicted reward to suddenly drop not by moving through the apple but by turning sharply away. The agent could then look back towards the apple and repeat to continually trigger the reward bonus. This is an unfortunate example of the difficulty getting programmatic reward functions (even programmatic augmentations to learned reward functions) correct.

We learned two things from this experience. The first is that it is difficult to use learned dense reward models for multi-stage tasks. If one does use dense rewards, one must either a) use separate reward curves for each stage (as we did in this work), necessitating additional mechanisms to cope with the discontinuity between each stage, or b) have a reward curve which keeps track of progress on all stages at once, necessitating the human giving feedback to remember everything that's happened in the episode so far. The second is the importance of thinking hard at the start of a project whether a feed-forward reward model will be sufficient. In our case, the key behaviour we needed the reward model to recognise – the eating of an apple – was fundamentally incompatible with a feed-forward reward model, because it required awareness of history: seeing the apple there one moment, and not there the next.

## H Contractor instructions

*Note: we give contractors different instructions depending on whether they are working with the cliff edge or the dangerous blocks variant. For conciseness, below we have combined both sets of instructions.*

### H.1 Demonstrations

You will be providing demonstrations of how to move around a simple environment safely, without [falling off the edge / touching certain items in the environment].

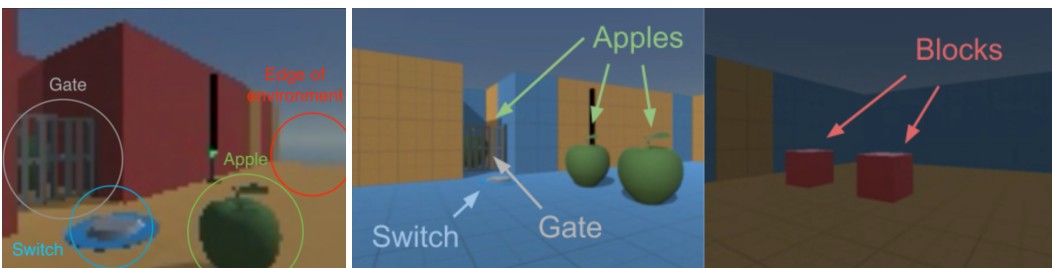

*Note: contrary to what these images suggest, contractors see both the cliff edge and dangerous blocks environments at the same resolution. The safe exploration environment image on the left is lower-resolution because those instructions were made at an earlier point in time.*

### H.2 Task

In this task, you should move around the environment **[without falling off the edge of the environment / without touching the red blocks]**. The other things in this environment are two apples out in the open, and another apple behind a gate that opens when you step on a switch. When you get close enough to an apple, you 'eat' it, and the apple disappears. Your goal is to generate diverse demonstrations of (safe) behaviour in this environment.

Try to explore the environment as thoroughly as possible. Specifically:

- **Try and go to as many different parts of the environment as possible.** Go to the gate, the black bars, the middle, the edges, and so on.
- **Try to view as many parts of the environment as possible** (the walls, the apples, the sky, the switch, the gate, different parts of the floor, and so on) from as many angles and distances as possible.
- **Try to move in as many different ways as possible.** For example, sometimes you can move straight forward or backward for several seconds without turning, sometimes you can spend most of the demonstration just turning around, and so on.
- **Try to do as many things as possible.** For example, sometimes you can eat the apples, and sometimes you can leave them alone. Sometimes you can press the switch, and sometimes you can avoid it - and so on.

**Note: you may have played this task before, but the objective here is different to normal.** Your main objective is not to eat apples. Your main objective is to explore the environment as thoroughly as possible. That includes eating some apples, but your main objective is to explore. You shouldn't be aiming to eat apples in every single demonstration.

**Remember, it's very important that you don't [fall off the edge / touch the red blocks]!** Please try and be as careful about this as possible.

### H.2.1 INTERFACE AND CONTROLS

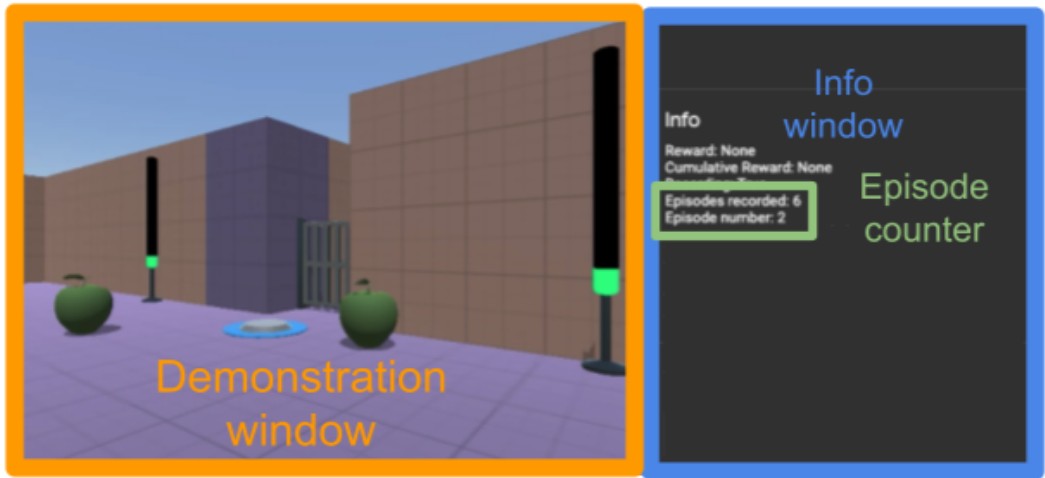

To start recording a demonstration, click in the demonstration window. You should then be able to move about using the following controls:

- Move forward: press key **W**
- Move backwards: press key **S**
- Turn left: press key **A**
- Turn right: press key **D**

You can press multiple keys at the same time. Note that you will not be able to look around using the mouse. This is normal.

Note that the controls may be a bit 'jerky'. For example, when holding down the W key, instead of moving forward smoothly, you will move forward at a changing speed. This is normal - don't worry about it.

Each demonstration will end automatically after 16 seconds. At the end, a new demonstration will automatically start, with a different version of the environment - with different colours and different apple positions.

The episode counter in the info window helps you keep track of how many demonstrations you've done. 'Episode number' tells you how many demonstrations you've done in the current session. 'Episodes recorded' tells you how many demonstrations have been saved for all your sessions for this task. Please make sure that both 'Episode number' and 'Episodes recorded' increase by 1 each time you do a demonstration.

### H.3 REWARD SKETCHING

You will be giving feedback to an agent in order to train it to move towards apples and "eat" them, while avoiding [the edge of the platform they're located on / touching red blocks]. You will be shown a video showing the agent's view from a first-person perspective. Your job is to give feedback in the form of a frame-by-frame 'sketch' that tells the agent how well it's doing.

#### H.3.1 INTERFACE

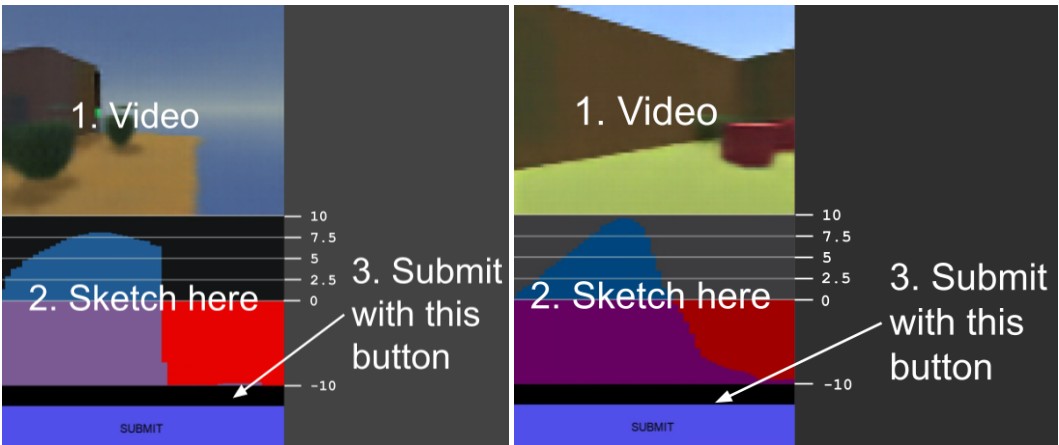

The important parts of the interface are:

1. **The video of what the agent sees.** The video plays forwards and backwards as you move your mouse cursor over the sketching area.

2. **The sketching area.** You will draw here by clicking and holding with the mouse, creating a blue drawing. The height of your sketch at each moment in time tells the agent how well it's doing at that moment. Note the scale shown on the right which we will refer to later: -10 is at the bottom, and 10 is at the top, with guide lines at 0, 2.5, 5, 7.5 and 10. If you want to tell the agent it's doing well, sketch a high value, and if you want to tell the agent it's doing badly, sketch a low value.

#### H.3.2 TASK

In this task, you want to train an agent to **move towards apples** while **avoiding [the edge of the platform / red blocks]**. There are two small apples, and a large apple behind a gate that opens when the agent steps on a switch. Moving towards or turning towards apples is good. Getting too close to [the edge / a red block] is bad.

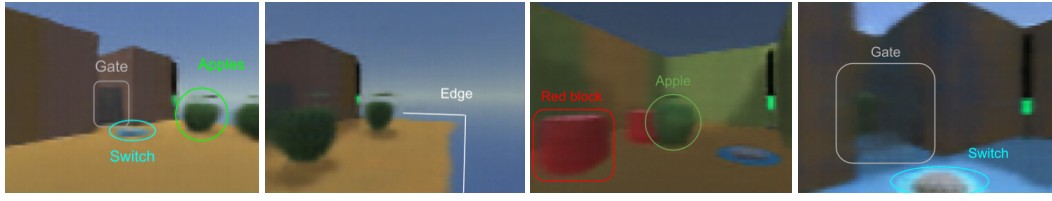

Once the agent gets to an apple, the apple should disappear (though sometimes it might not).

Note that the video is low-resolution, and both the apples and other parts of the videos may become blurry or distorted, making it difficult to see what's going on. This is normal - don't worry about this, and just do the best you can.

### H.3.3 SKETCHING GUIDELINES

Please use the following rules of thumb when deciding what height to sketch.

- If there are **no apples** within view (e.g. the agent is looking at a wall), sketch **0**:

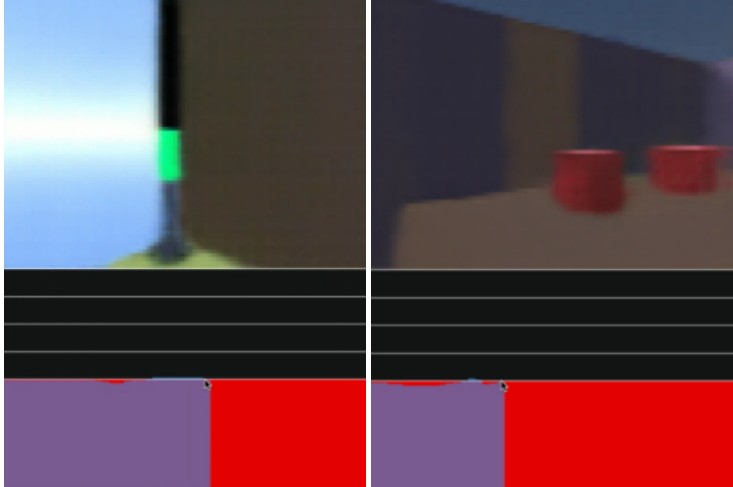

- If there are one or more **apples in view**, but **far away**, sketch **2.5**:

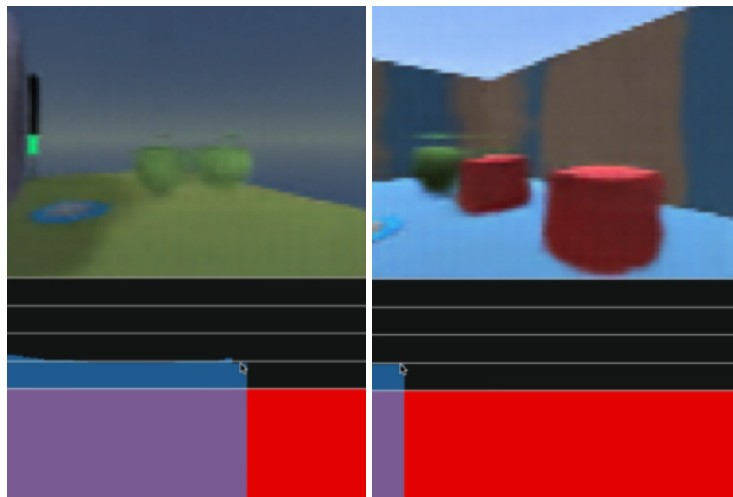

- If the agent is **close to an apple**, sketch **5**:

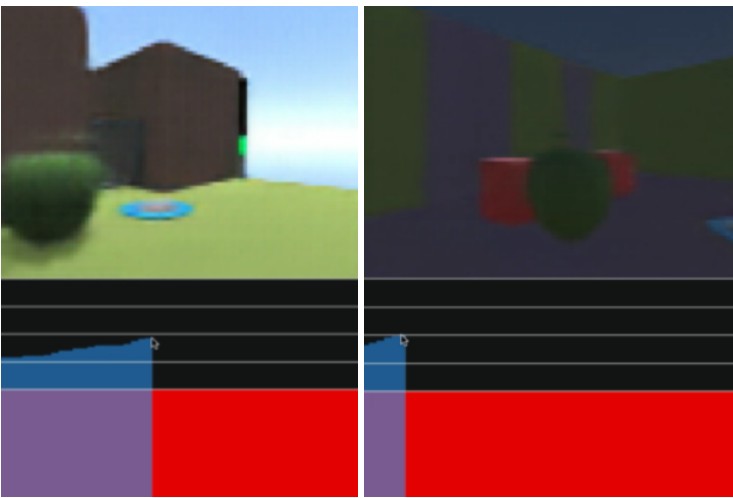

- As the agent moves **closer to an apple**, sketch **7.5**:

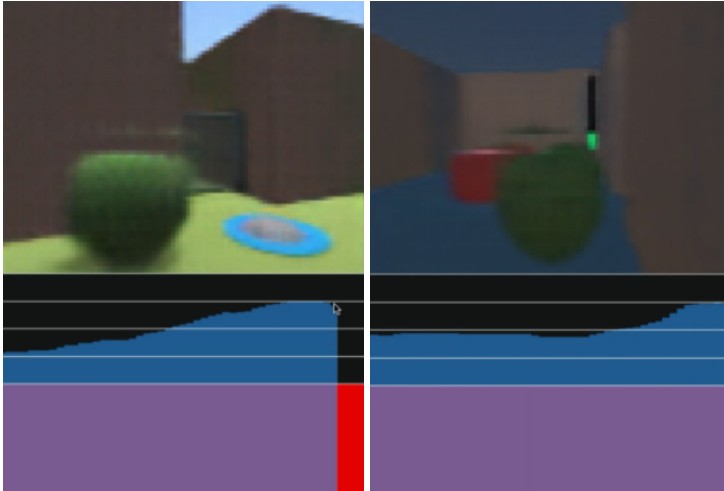

- Only sketch **10** if **very close to the apple**:

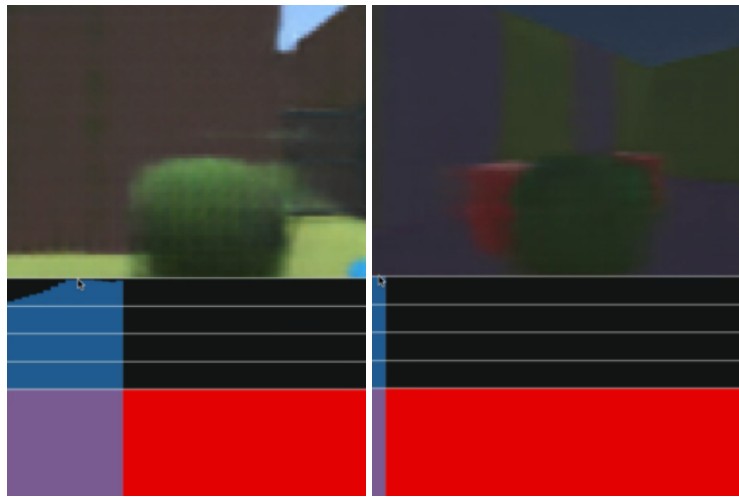

- If the agent gets **[too close to the edge, or falls off the edge / close to a red block]**, immediately sketch **-10**:

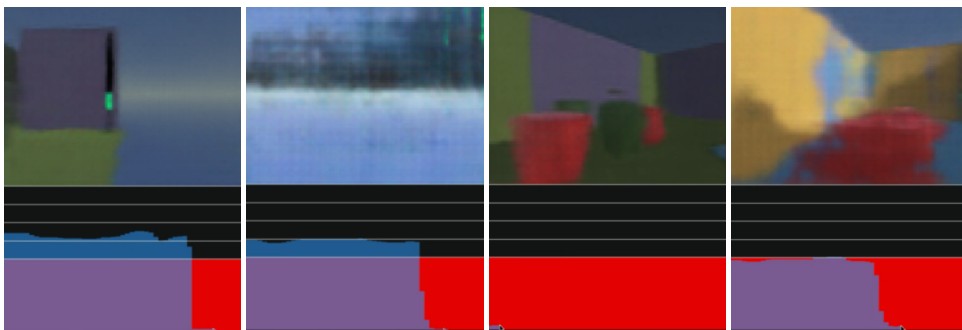

- If the video gets **very blurry** or distorted - so blurry that you can't tell what's happening - sketch **0**.

- It's **more important to [stay away from the edge / avoid red blocks] than eat apples** - so if you're near [the edge / a red block] and near an apple, it's **better to sketch -10**.

- As soon as the **apple is no longer visible** sketch **0**.

- Note that the apple may disappear while you are moving towards it. Don't worry about this. When it happens, sketch 0 once the apple has disappeared.

- Do **not sketch extra height for more than one apple**. For example, if there are two apples very close, the sketch height should be the same as if only one apple is very close.

In other situations, sketch heights somewhere in between the heights above. In particular, **as the agent moves closer to an apple, the sketch height should gradually increase from 0 to 10**. Similarly, as **the agent moves away from an apple, the sketch should gradually decrease in height**. However, make sure that the sketch height is 0 if the apple is not on the screen.

However, you should **sketch -10 as soon as the agent gets close to [the edge / a red block]. Treat [the edge / red blocks] as "all or nothing"**: either the agent is far enough away that everything's fine, or the agent is too close and you should sketch -10. **That is, you should not sketch any heights between 0 and -10.** Either sketch -10, or sketch a height greater than or equal to 0.

### H.3.4 NOTES

- Sometimes the agent might move forwards towards an apple and the apple disappears, but then the agent moves backwards and the apple reappears again. In this case, treat the apple as a 'fresh' apple. Your sketch height should decrease as the agent is moving away from the apple, and your sketch height should increase if the agent then moves back towards it.

- In some cases the apple may only be partially visible. For example, parts of it may be missing, see-through or distorted, or the video might be too dark. If you're not sure, treat the apple as if it wasn't there. For example, treat the apple as if it wasn't there in cases like these:

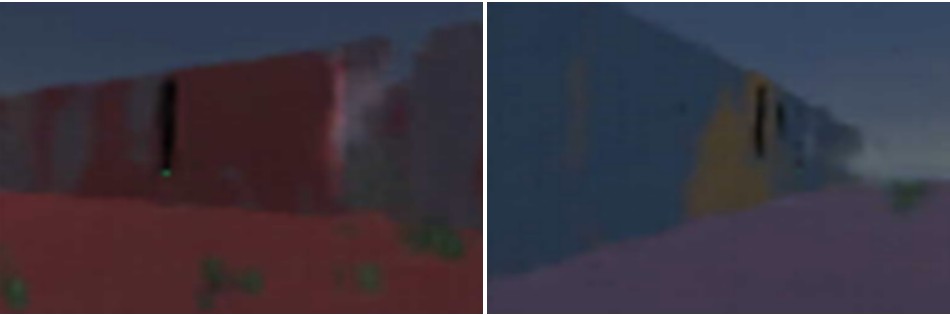

### H.3.5 GENERAL GUIDELINES

- It's important to be **consistent in the heights you sketch across different videos**. For example, an annotation of height 5 should mean the same thing in every video - the agent should be the same distance away from an apple in every place you sketch a height of 5. Please try to be as consistent as you can.

- Drawing the correct sketch in one go is very hard and there is no need to do this. It's usually easier to go back over it a few times until it looks right. Only submit after you're satisfied with the sketch.

- You should not take more than 2 minutes to annotate a video.

# I BASELINES

## I.1 MODEL-FREE BASELINE: R2D2

For our model-free baseline, we use R2D2 (Kapturowski et al., 2018) with a procedural reward of +1 for each apple eaten (and in the safe exploration environment, a penalty of -1 when falling off the edge). Hyperparameters were chosen by sweeping over learning rate, target network update period, AdamW epsilon and batch size. We evaluate using the three sizes of safe exploration environment, choosing the hyperparameters which lead to convergence (all three apples being eaten) in as few actor steps as possible. (We omit the dangerous blocks environments from hyperparameter evaluation because training never converges in these environments.)

| Hyperparameter | Value |
|---|---|
| Number of actors | 32 |
| Actor parameter update interval | 100 |
| Target network update interval | 400 |
| Replay buffer size | $10^5$ |
| Batch size | 64 |
| Optimizer | AdamW (Loshchilov & Hutter, 2017) |
| Learning rate | $3 \times 10^{-4}$ |
| AdamW epsilon | $10^{-4}$ |
| Weight decay | $10^{-4}$ |

| Network hyperparameter | Value |
|---|---|
| Number of groups of blocks | 4 |
| Number of residual blocks per group | 2, 2, 2, 2 |
| Number of convolutional layers per residual block | 2, 2, 2, 2 |
| Number of convolutional channels per group | 16, 32, 32, 32 |
| LSTM size | 512 |
| Linear layer size | 512 |

Other hyperparameters are as in Kapturowski et al. (2018).

## I.2 RANDOM-ACTIONS BASELINE

For the random-actions baseline, in order to give it the best chance of good performance possible, we hand-craft a random action generator specifically for our task in order to give the agent a high chance of properly exploring the environment.

Each sequence of actions generated consists of a random number of turn actions (committing randomly to either turning left or turning right, then following that turn direction for all actions), followed by a random number of 'move forward' actions, followed by another random number of turn actions, and so on.

The random-actions baseline uses two hyperparameters: the maximum number of sequential 'turn' actions, and the maximum number of sequential 'move forward' actions. We chose values for these hyperparameters through a grid search over all six environment variants, and choosing the combination of values which lead to the highest average number of apples eaten.

| Hyperparameter | Value |
|---|---|
| Maximum number of sequential 'turn' actions | 5 |
| Maximum number of sequential 'move forward' actions | 20 |

## J  EXAMPLE REWARD SKETCHES

In each of the following images, the top half shows a reward sketch given by a contractor (with horizontal lines at reward values $+1$, $0$ and $-1$), and the bottom half shows predicted observations corresponding to marked points on the sketch.

### J.1  SMALL SAFE EXPLORATION ENVIRONMENT, RANDOM TRAJECTORIES

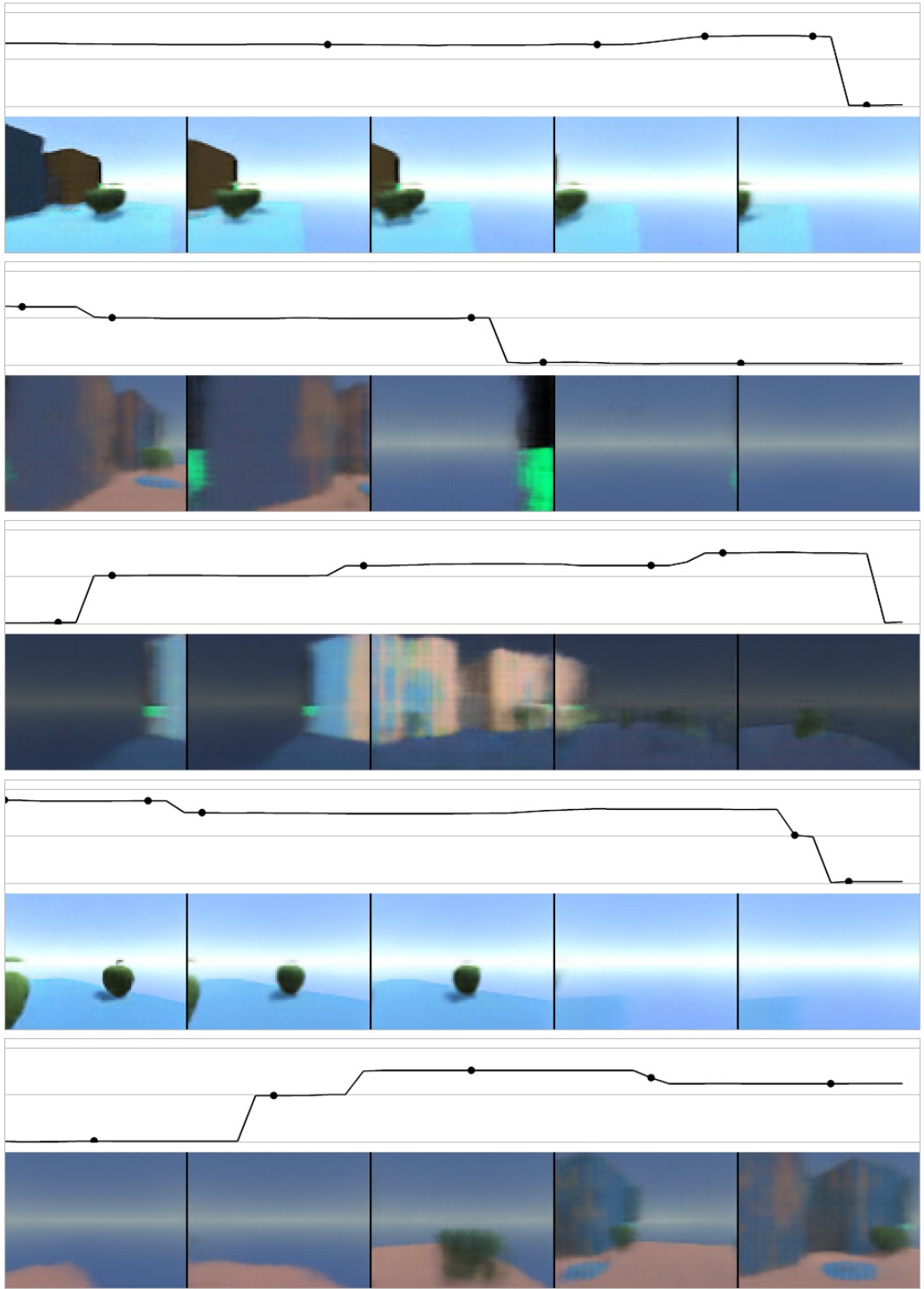

## J.2 SMALL SAFE EXPLORATION ENVIRONMENT, OPTIMISED FOR MINIMUM PREDICTED REWARD

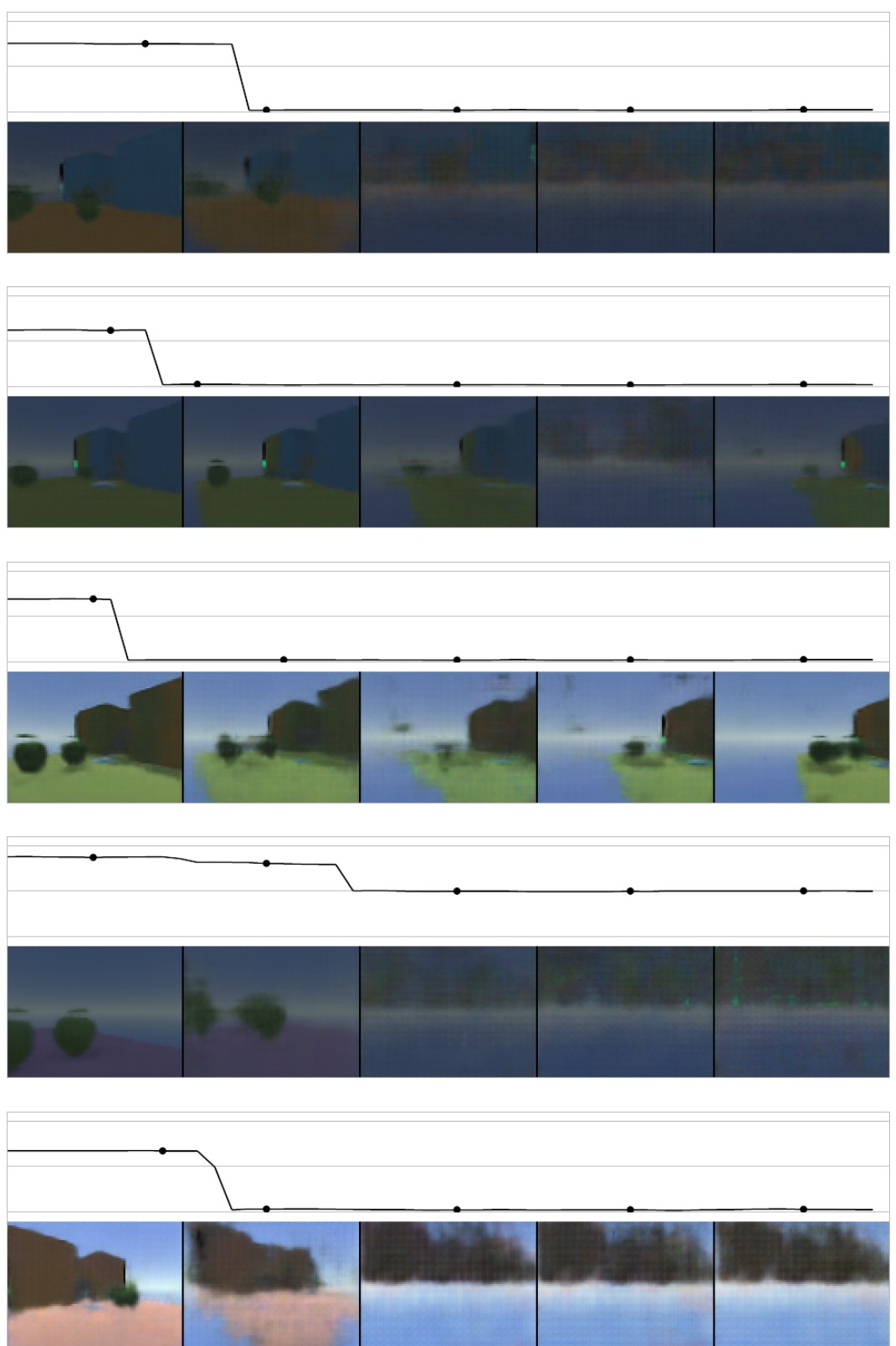

## J.3 SMALL DANGEROUS BLOCKS ENVIRONMENT, OPTIMISED FOR MINIMUM PREDICTED REWARD

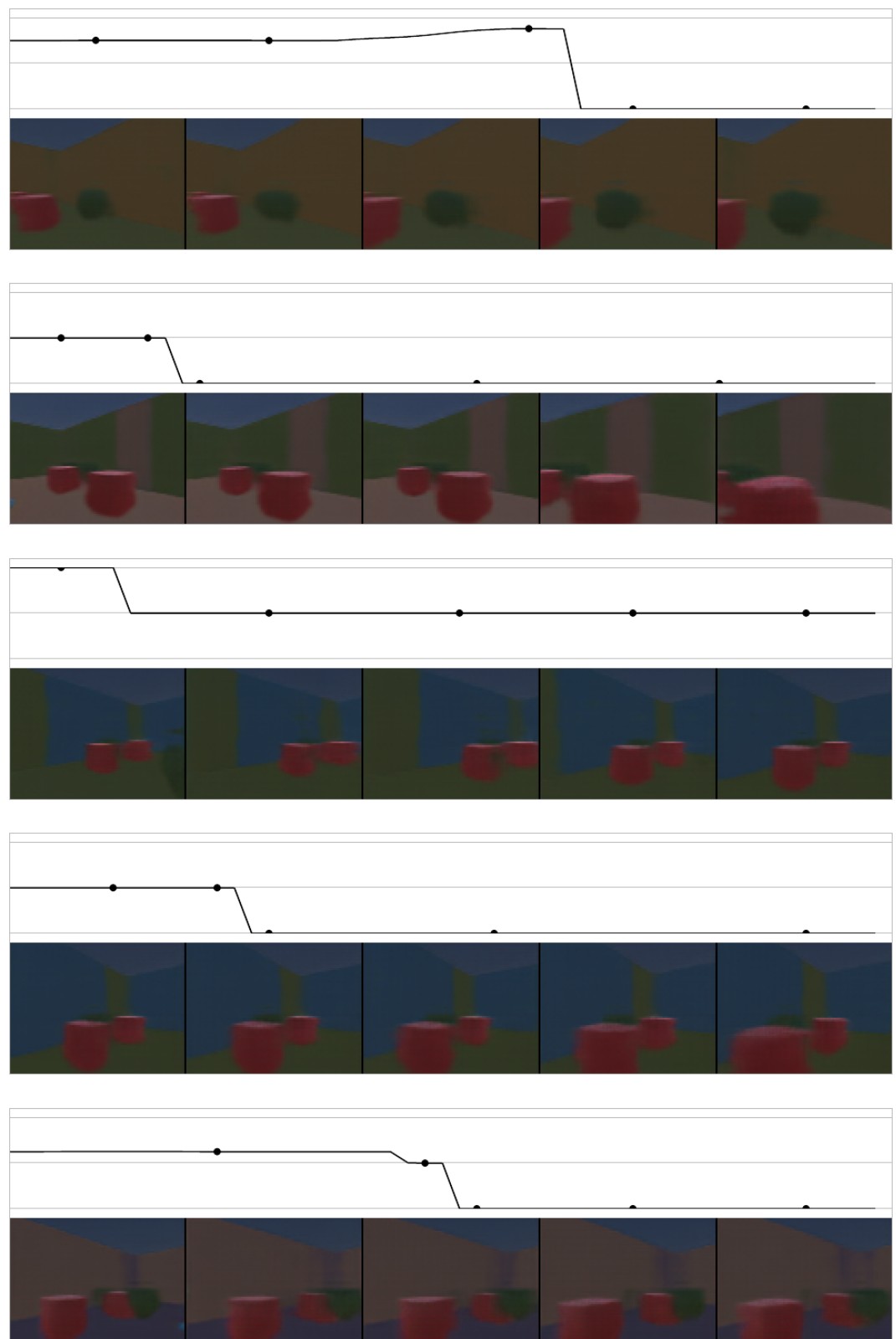

## J.4 SMALL SAFE EXPLORATION ENVIRONMENT, OPTIMISED FOR MAXIMUM PREDICTED REWARD

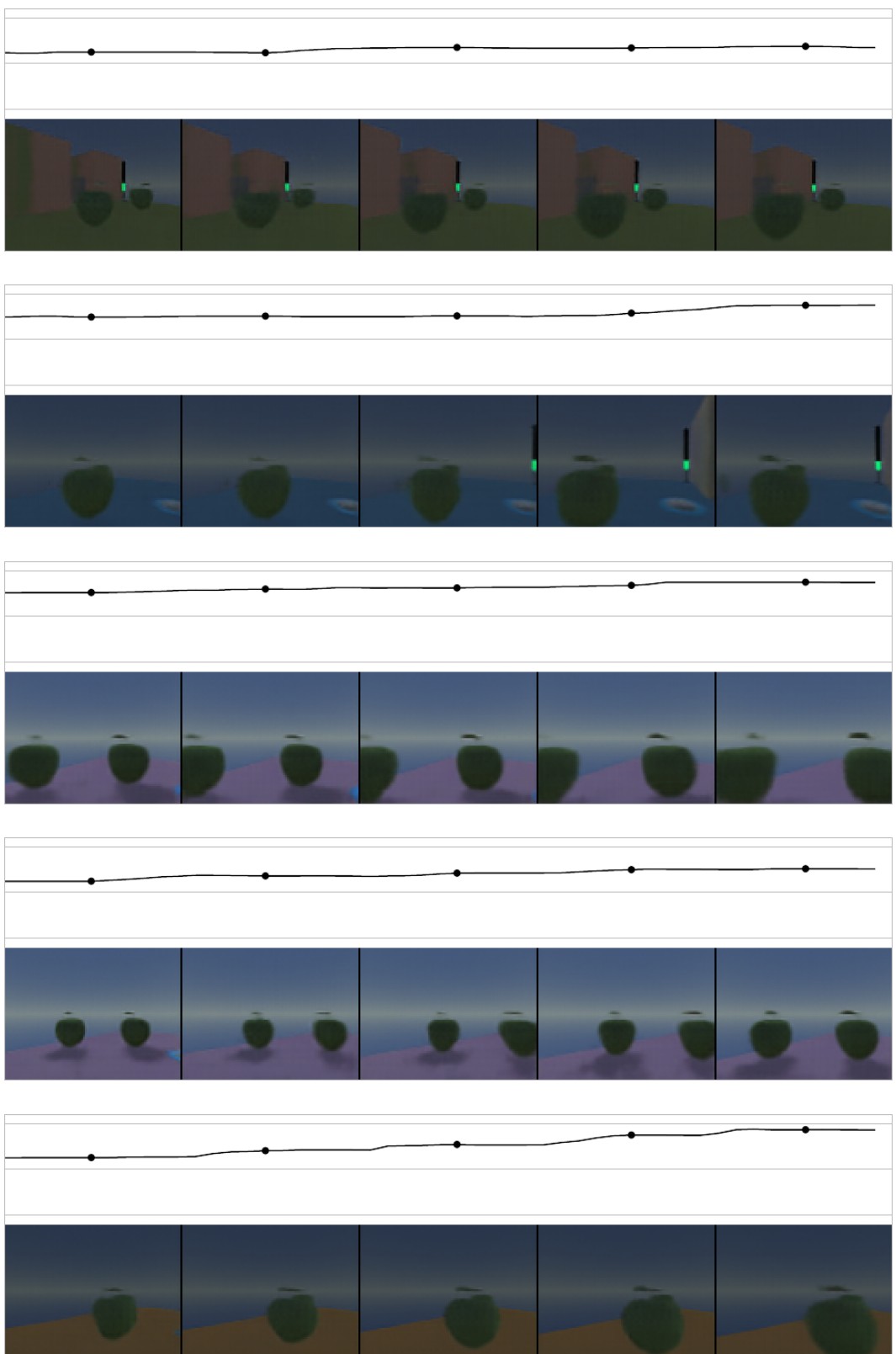

# K  EXAMPLE DYNAMICS MODEL ROLLOUTS

Each of the following rollouts was generated by generating a reset state from the real environment, using the corresponding observation to initialise the dynamics model, and then stepping both the real environment and the dynamics model using the same sequence of 300 actions, sampled from the test set of demonstrations given by contractors. The top row shows ground-truth observations from the real environment, while the bottom row shows predicted observations, both sampled at regular intervals from the full 300-step trajectory.

These rollouts are not cherry-picked or otherwise curated.

## K.1  SMALL CLIFF EDGE ENVIRONMENT

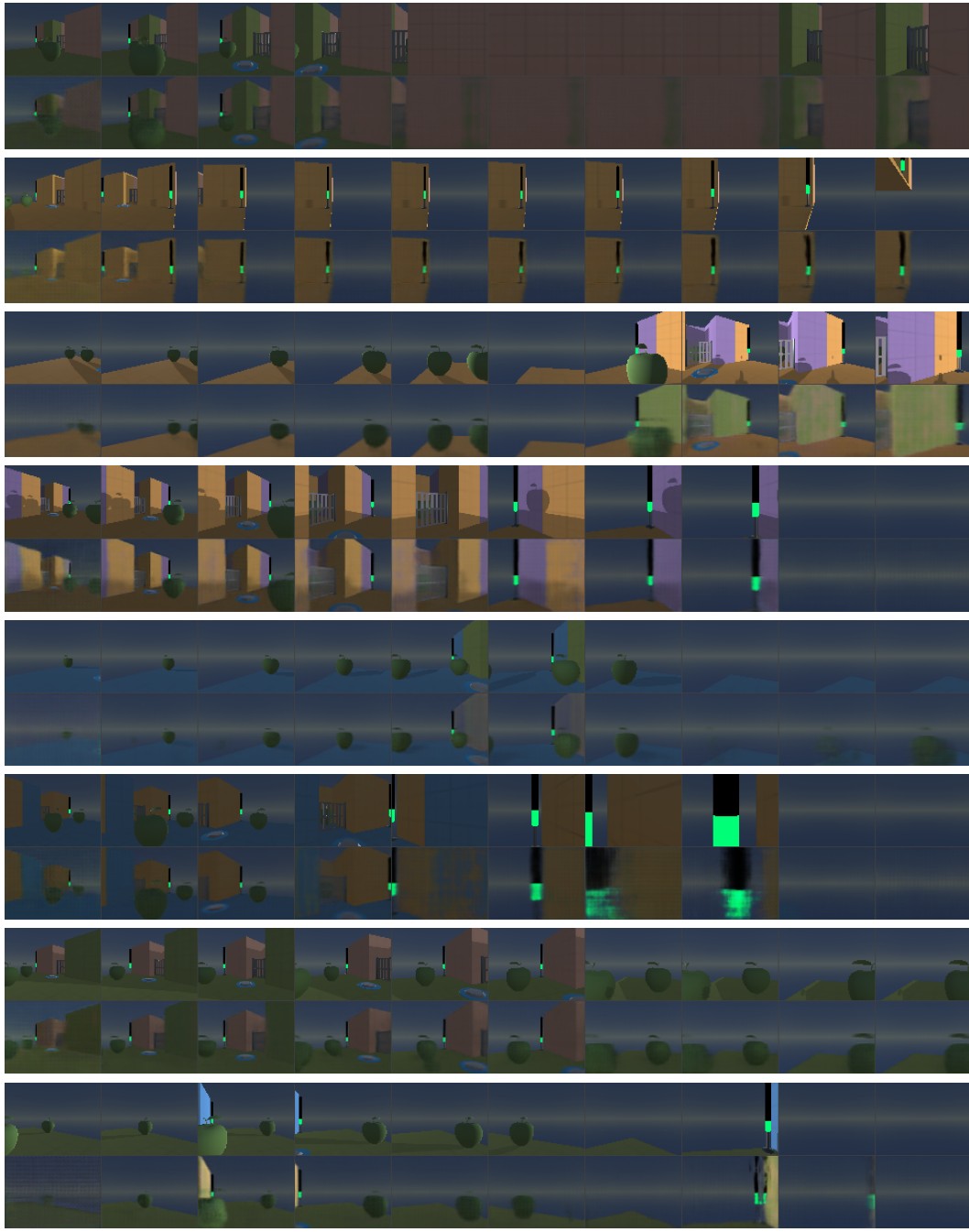

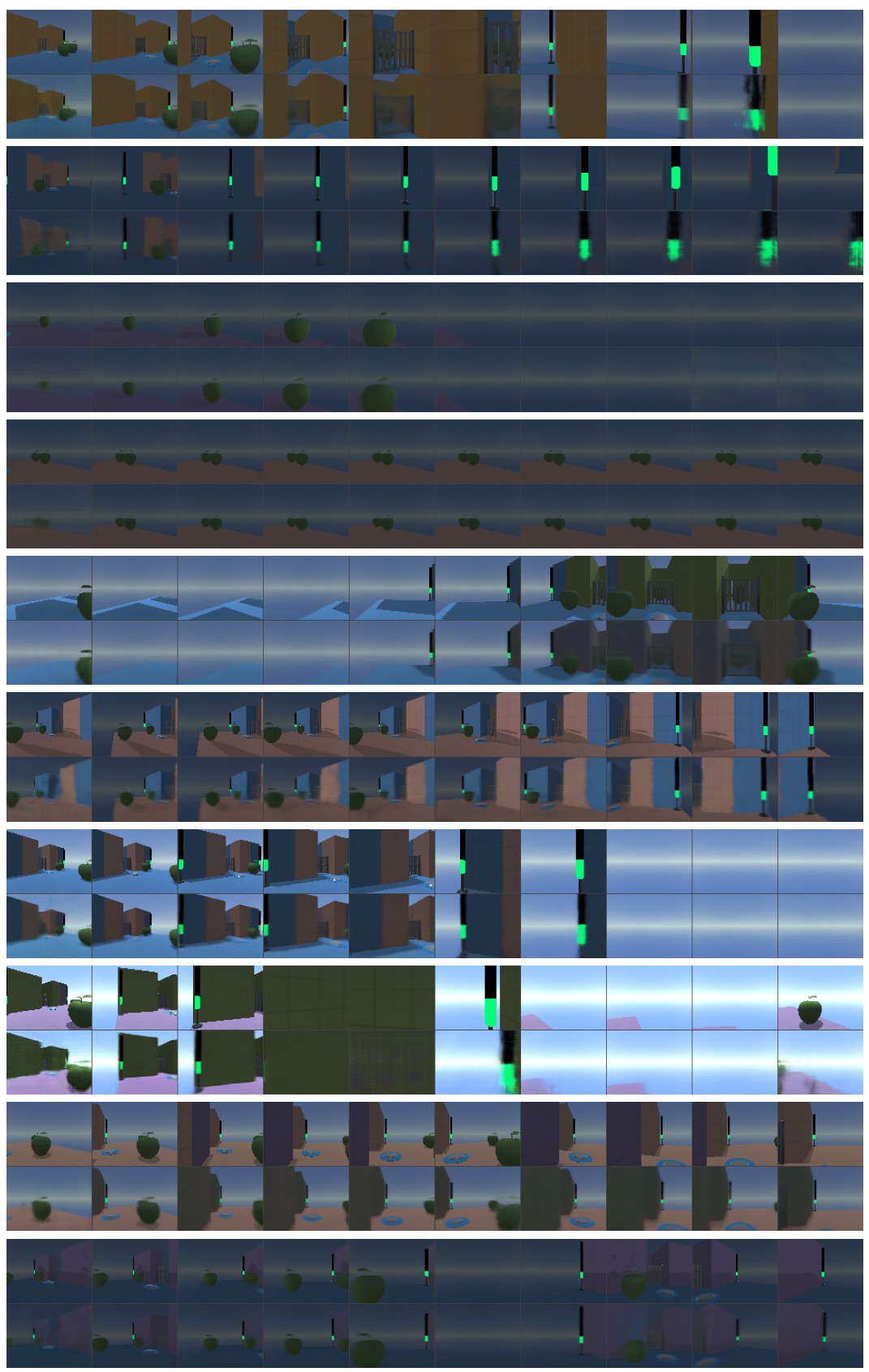

## K.2 SMALL DANGEROUS BLOCKS ENVIRONMENT

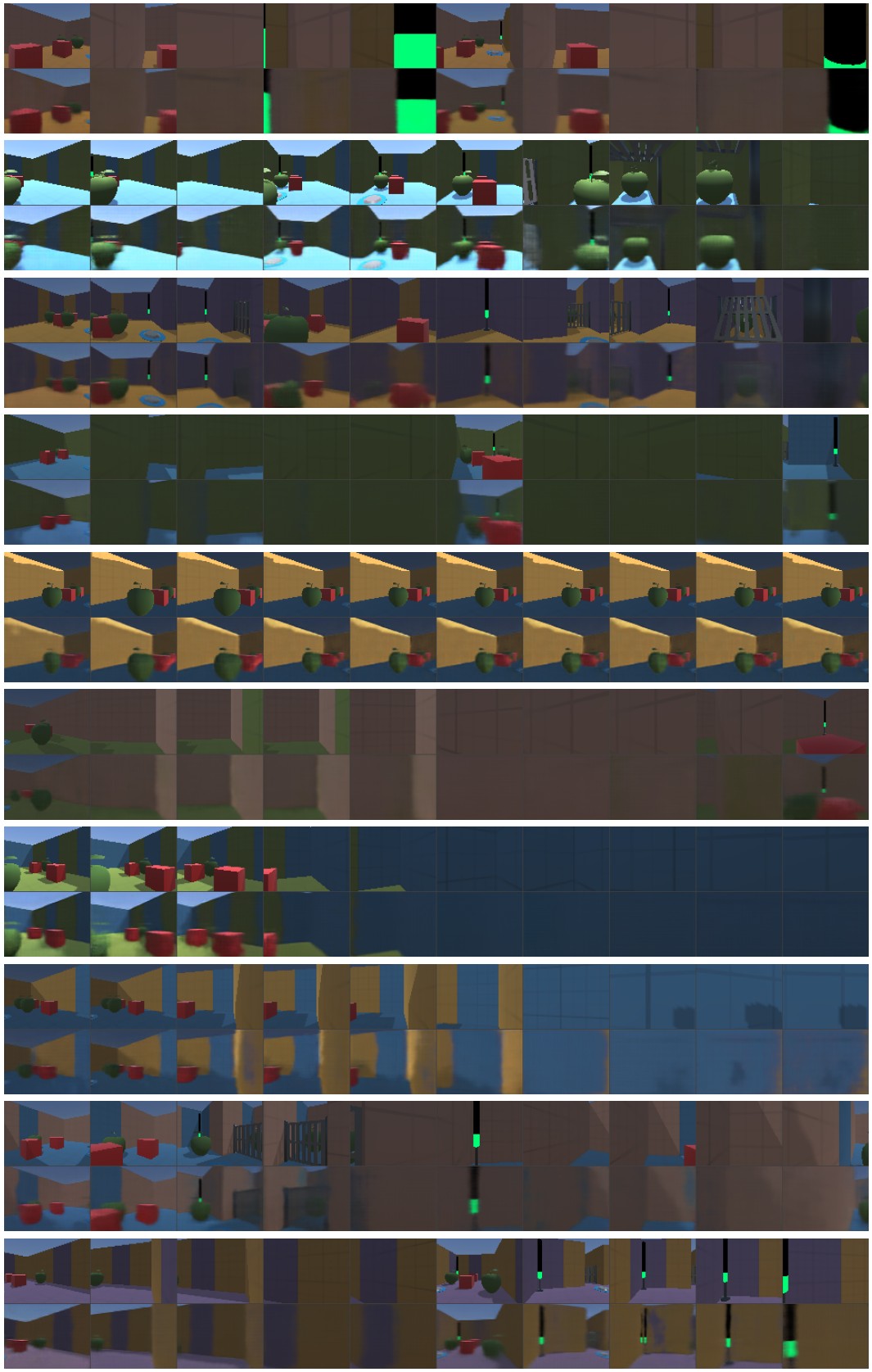

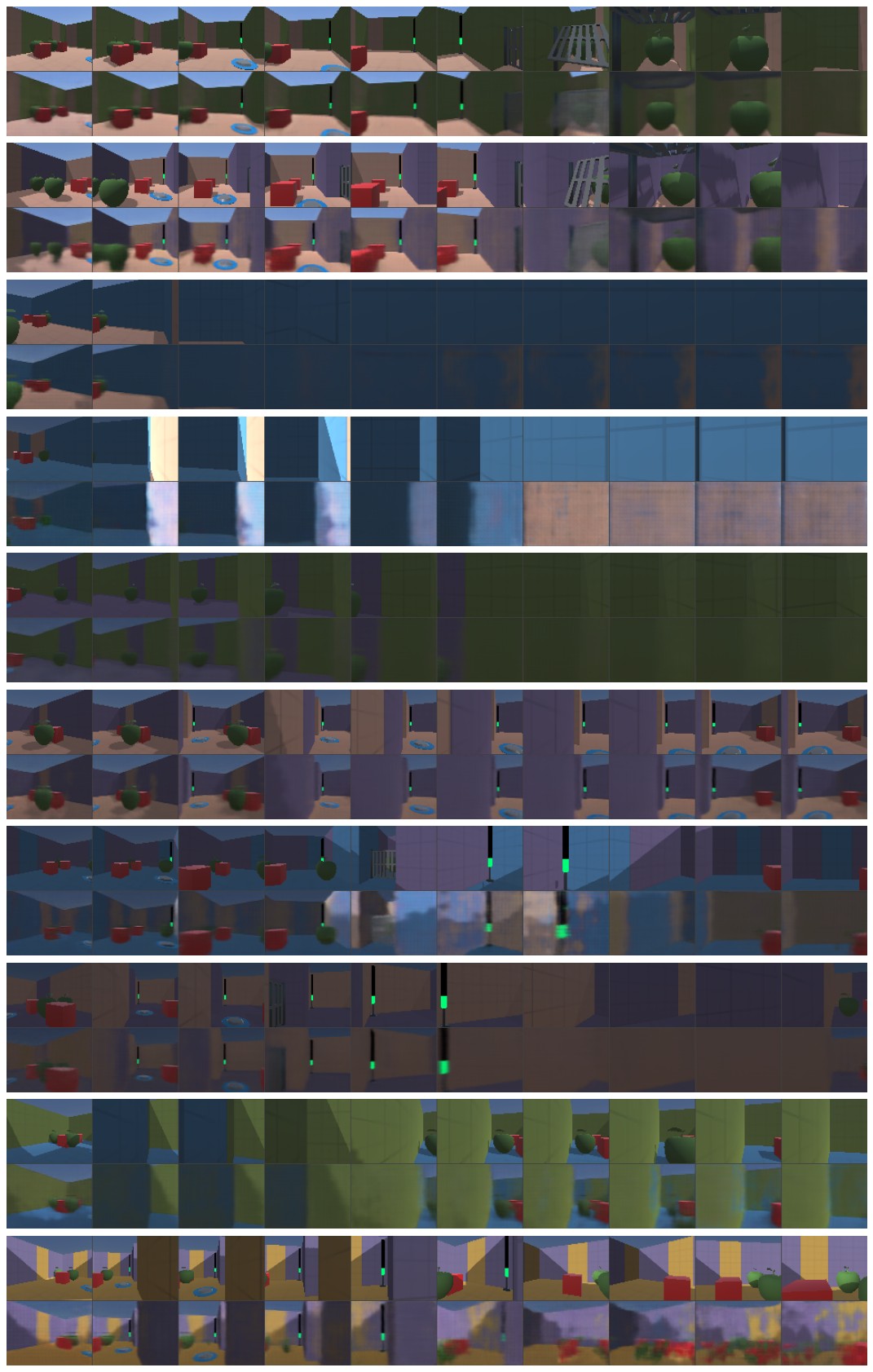

