# OpenReview forum: "Safe Deep RL in 3D Environments using Human Feedback"
_ICLR.cc/2022/Conference — ICLR 2022 Submitted_

### Official Review · Reviewer_yf75 · 2021-10-28

**Correctness:** 4
**Technical Novelty And Significance:** 1
**Empirical Novelty And Significance:** 2
**Recommendation:** 3
**Confidence:** 4

**Main Review:**

I generally like the idea of the paper to bring ReQueST together with human feedback through reward sketches. The paper is well-written and easy to follow. The experimental setup is also well-defined and described. However, the second environment should be better justified as the setting looks very unusual.
The authors clearly state their contributions and delineate it from previous work. The use of real human data and reward sketches is laudable, and an important contribution. Experiments are replicated 10 times, which is a sensible amount.

However, this only seems to be a very incremental work over the existing algorithm. Additionally benchmarking ReQueST on 3D environments is valuable but not a contribution that is sufficient for publication in ICLR. The paper is better suited as a workshop contribution.

Also, I'm not convinced about the numerical results. Their main safety result shows that ReQuest is unable to outperform a simple baseline (more on this baseline in a bit) during the evaluation, performing less safe and gathering fewer apples in the "Cliff Edge" environment. The authors should identify the problems encountered by ReQuest and their modifications here and propose a solution that improves the performance to be better than the baseline. I would also actually argue that having constraint violations in the gathered dataset should improve the performance for an algorithm (not necessarily for ReQuest), since the agent should exploit the knowledge on constraints this dataset contains.

Some further points:
-	In Fig. 2 you could more explicitly show the difference to original ReQueST
-	While the authors use a large amount of replications, they only include a classical RL baseline. This baseline uses a simplistic, sparse reward, rather then the richer reward sketches used by ReQuest. Also, it penalizes constraint violations by the same amount that it rewards successes. This kind of baseline method will never perform safely during training.
-	The paper should also compare itself against well-known SafeRL algos and/or against classifier-based approaches (such as the original version, why don’t you compare to vanilla-ReQueST in order to show the benefits?). Also, while the authors argue against imitation and offline learning, such approaches could also serve as baselines and provide experimental evidence.
-	Since they claim to provide superior safety, the authors should compare their method to recent Safe RL methods, like 2020 Stooke: Responsive Safety in Reinforcement Learning by PID Lagrangian Methods, or 2020 Srinivasan: Learning to be Safe: Deep RL with a Safety Critic. The latter method uses a pretraining step to bootstrap the safety critic before training, which could use a similar dataset than the one used in this study. Rather than only evaluating ReQuest, it could be a more interesting paper if multiple Safe RL approaches based on the dataset are evaluated.


**Summary Of The Paper:**

The paper proposes an extension to ReQueST, which learn learns a neural simulator of the environment from safe human trajectories and then learns a reward model from human feedback. This work extends ReQueST by dense reward sketches on imagines trajectories and evaluates the idea on a visually-complex 3D environment. The paper also discusses the amount of training data that is needed for ReQueST to work in such a setting. The authors collect 160 person-hours of "safe" human exploratory trajectories and 10 person-hours of reward sketches. Their results show that the application of ReQuest results in "3 to 20" times less constraint violations compared to a non-safe baseline.

**Summary Of The Review:**

I generally like the idea of the original paper (and hence also this paper) but in my opinion this paper is not enough for publication in ICLR. The contribution is very limited over the original ReQueST.

---

> ### Author Response · Authors · 2021-11-16
> **Reply to Reviewer yf75**
>
> Thank you for your thorough review, and for your kind words on the good things about the paper!
>
> If we understand correctly, the main crux for you is the apparent lack of novelty. We're surprised at this assessment; we hoped that using real human data, a richer form of feedback (reward sketches rather than reward labels), and significantly more challenging and realistic environments would be regarded as large enough improvements over the original work on ReQueST to clear the novelty bar here. Would you be able to suggest any examples of ways we could make this work non-incremental?
>
> This main crux aside, below we address individual points.
>
> On the numerical results: we agree it's not ideal that the ReQueST agent falls off the edge 7 times during evaluation in the hard cliff edge environment, and that it doesn't manage to eat all 3 apples in any environment variant. However, note that our research question places just as much weight on train-time metrics as evaluation-time metrics - and taking that broader view, it is clear that the ReQueST agent does outperform the baseline on the key metric of interest (number of safety violations), while still achieving moderate performance.
>
> Having said that, you're right to say we could do a better job of talking about the problems encountered by the ReQueST agent. We have some discussion of failure modes on the website, but we can certainly do a better job of making this information available in the paper itself, and explaining the specific failures seen in the numerical results. We'll add an appendix detailing a) what causes the ReQueST agent's safety violations, b) why ReQueST does not always collect all three apples, and c) our methodology for determining the answers to these questions (which is itself somewhat involved).
>
> On comparison to other approaches: for safe RL algorithms, see our paper-level comment; on comparison to vanilla ReQueST (that is, using sparse reward feedback instead of dense reward sketches), we are in fact running these experiments now and hope to add results in the week.
>
> On additional baselines (imitation learning, offline RL): these would indeed be interesting sources of information; we probably won't have time to include them in our current submission, but will bear this in mind for future work.
>
> Finally, on penalising constraint violations by the same amount as rewarding successes for the model-free baseline: we *think* it was the case that setting the penalty too high resulted in training not converging. We'll double-check this and get back to you.

---

> > ### Comment · Reviewer_yf75 · 2021-11-21
> > **Reply to author's response**
> >
> > > If we understand correctly, the main crux for you is the apparent lack of novelty. We're surprised at this assessment; we hoped that using real human data, a richer form of feedback (reward sketches rather than reward labels), and significantly more challenging and realistic environments would be regarded as large enough improvements over the original work on ReQueST to clear the novelty bar here. Would you be able to suggest any examples of ways we could make this work non-incremental?
> >
> > The paper proposes two contributions: (1) it extends ReQueST to work on 3D environments, and (2) it applies human feedback through reward sketches. However, (1) ReQueST has already worked on simple navigation task and 2D racing car. I think the paper did not made any significant changes to ReQueST to make it work for 3D environment. Taking an existing algorithm/method and running on difficult problem seems incremental work to me. (2) Also reward sketches are nothing new. They have been applied in a variety of settings and papers. Optimizing imagined trajectories has some novelty in this scenario but not at a level that I would suggest acceptance at ICLR.
> >
> > >On comparison to other approaches: for safe RL algorithms, see our paper-level comment; on comparison to vanilla ReQueST (that is, using sparse reward feedback instead of dense reward sketches), we are in fact running these experiments now and hope to add results in the week.
> >
> > >On additional baselines (imitation learning, offline RL): these would indeed be interesting sources of information; we probably won't have time to include them in our current submission, but will bear this in mind for future work.
> >
> > The paper's originality and significance would benefit from an experimental evaluation against the state of the art. I understand that evaluating the proposed approach against the mentioned state of the art is anything but trivial.
> >
> > That being said I see a nice piece of work here that is interesting to anyone who wants to apply ReQueST-like approaches to high-dimensional state spaces and who might want to resort to human data. But I must say that I still only see incremental work over existing literature that is - in my opinion - better suited for a workshop contribution (where it might also benefit from and improve upon).
> >
> > BTW: I do not see a new revision of the paper. Only those from Sep. 29th.

---

> > > ### Author Response · Authors · 2021-11-23
> > > **Reply to Reviewer yf75's reply**
> > >
> > > > Taking an existing algorithm/method and running on difficult problem seems incremental work to me.
> > >
> > > Alright, fair enough. Thanks for the feedback on this point!
> > >
> > > > The paper's originality and significance would benefit from an experimental evaluation against the state of the art. I understand that evaluating the proposed approach against the mentioned state of the art is anything but trivial.
> > >
> > > Understood. We'll consider this for our future work.
> > >
> > > > BTW: I do not see a new revision of the paper. Only those from Sep. 29th.
> > >
> > > Strange. We've double-checked and it's definitely visible from our side - both through the main 'PDF' link at the top of the page, and in the 'Show Revisions' link (though on that page it's not very visible - it's the "Blind Submission by Conference" entry at the top). Figures 5 and 6 should have an extra bar, 'ReQueST (sparse)'. Does the change definitely not show up for you in either of those two places?

---

### Official Review · Reviewer_gqNf · 2021-10-30

**Correctness:** 3
**Technical Novelty And Significance:** 2
**Empirical Novelty And Significance:** 2
**Recommendation:** 5
**Confidence:** 3

**Main Review:**

The paper shows that ReQueST is plausibly a general purpose solution to  the safe exploration problem: where safe behaviour is learned from humans, rather than given by a procedural function, and the simulator can be learned from data.

There are several weakness, for example, the task of apple picking, is this sufficient to show the challenge of RL or more difficult tasks should be used?

The paper also does not do a detailed comparison with existing approaches. If no existing approaches can achieve what the paper proposed to do, then please clearly say so. Otherwise please do a comparison to strengthen the  claims of novelty of the paper.

**Summary Of The Paper:**

This paper shows that  ReQueST can be used to train an agent in a 3D environment with an order-of-magnitude reduction in instances of unsafe behaviour than typically required with reinforcement learning. No procedural specification of safe behaviour is required with minimal assumptions other than unsafe or near-unsafe states being recognisable by humans.



**Summary Of The Review:**

Overall it is an interesting paper with some good results shown. I am not sure the results are sufficient enough. I would also like to see more detailed comparisons with the state-of-the-art.

---

> ### Author Response · Authors · 2021-11-16
> **Reply to Reviewer gqNf**
>
> Thank you for your thorough review!
>
> For comparisons to existing approaches, please take a look at our paper-level comment.
>
> Addressing the question of whether the task is sufficiently difficult: note that the task involves 3D navigation from pixels (a partially-observable and very large state space), multiple apples, obstacles to avoid, and a sensor being used to open a gate. Although the environment is simpler than those typically used to test the performance of deep RL algorithms, note that safety is the metric of interest in this work - and a) our environment is already challenging in terms of safety for our baseline deep RL algorithm, and b) our environment is significantly more challenging than the kinds of environments typically used for safe RL research (e.g. https://openai.com/blog/safety-gym).
>
> Overall, on whether the results are "sufficient enough", would you be able to give us any more details on what further results you'd like to see?

---

### Official Review · Reviewer_rmJm · 2021-11-02

**Correctness:** 4
**Technical Novelty And Significance:** 2
**Empirical Novelty And Significance:** 3
**Recommendation:** 5
**Confidence:** 3

**Main Review:**


### Concerns
1. This method ensures safety by learning the dynamics of safe human demonstrations. However, if some of the human demonstrations are unsafe, would there be a way to know? Or would this method be sensitive to those cases?
2. This method requires a lot of human data, which is ok considering the challenge of the task. One concern is that it seems unclear how to determine when the training for dynamics (and reward) are "completed." Especially for safety-critical tasks like the ones suggested by the authors, what would be the indicators in the proposed method that can suggest: "ok, now the learned model is safe enough, please feel free to deploy."
3. A programmatic reward bonus is applied to the proposed method, not baselines. Would this mean that the reward learning component in this method is insufficient to recover the reward function?

### Minor points
1. Env2 has blocks in the world, where if the agent moves the block, the reward it will receive will change. This is the main reason why in the results, model-free RL performs much worse than the proposed method.
    * I think this is a good example of reward hacking, where the defined reward - apple picking and avoiding touching blocks, does not match the actual goal of the task - apple picking. Hence, model-free RL performed poorly. The proposed method leverage human demonstration to avoid suffering from the misspecified reward.
    * This indicates the power of the proposed method to handle misaligned rewards. However meanwhile, this might also indicate that the proposed method cannot completely outperform model-free RL when the reward is specified well, such as in the cliff env.


**Summary Of The Paper:**


### Contributions
* This paper proposes a safe model-based deep RL approach where
    * No simulator is needed. The dynamic model is learned from data.
    * No constraint is specified.
* This work is an extension of reward query synthesis via trajectory optimization (ReQueST).

* ReQueSt (previous work)
    * Algorithm
        * Dynamics learned from (potentially unsafe) random exploration.
        * Reward learned from binary human feedback that is generated by a procedural reward function.
        * Demonstrated in State-based 2D navigation (non-pixel-based), Image-based Car Racing (pixel-based, 64x64×3).
    * Regarding safety
        * ReQueST avoids the problem of safe exploration by allowing agents to explore these states in a simulated model without having to visit them in the real environment.
* This work
    * Algorithm
        * Dynamics learned from safe demonstration provided by humans (160 person-hours).
          * Similar to ReQueSt, `except that we use a larger encoder network, a deconvolutional decoder network, and train using a simple mean-squared error loss between ground-truth pixel observations and predicted pixel observations.`
        * Reward learned from
            * Type of feedback from humans: reward sketch (previous work) (10 person-hours).
            * The proxies for value of information: maximization and minimization of reward as predicted by the current reward model.
            * The algorithm for reward learning: MPC.
        * Demonstrated in 3D env
          * Task goal = eat all the 3 apples
          * Env 1 = Cliff edge environment
            * Safety = not fall off the end of the world
          * Env 2 = Dangerous blocks environment
            * Safety = bumping into blocks
          * Each env has 3 Sized subvariants.
          * Each env is pixel-based, 96×72×3.
    * Regarding safety
        * Safety is achieved by focusing the learned dynamic model in the safe scenarios, with only the safe data. `Given models of sufficient fidelity, this should allow us to train an agent with close to zero instances of unsafe behaviour in the real environment.`

### Results
* Env 1 = Cliff edge environment (Fig5)
    * Safety violations during training: proposed method << model-free RL
    * Safety violations during testing: proposed method = model-free RL << random policy
    * Apples eaten: random policy <= proposed method < model-free RL
    * => Proposed method vs model-free RL => tradeoff between safety violations during training and apples eaten
* Env 2 = Dangerous blocks environment (Fig6)
    * Safety violations during training: Proposed method << model-free RL
    * Safety violations during testing: Proposed method << model-free RL = random policy
    * Apples eaten: random policy = model-free RL < proposed method
    * => Proposed method is better than model-free RL and random


**Summary Of The Review:**

This paper is an extension of the previous work ReQueSt, with the focus on learning safe policy from human demonstrations only, without simulators or specifications. There are some concerns about the approach. It has a very nice demonstration in challenging pixel-based 3D tasks.

---

> ### Author Response · Authors · 2021-11-16
> **Reply to Reviewer rmJm**
>
> Thank you for your detailed review and interesting questions!
>
> First, addressing the concern about the **safety of human demonstrations**. The direct answer to the question "Would there be a way to know if some demonstrations are unsafe?" is that we would expect the safety violations to be obvious while collecting the demonstrations - e.g. physical damage being caused to the agent or the environment. However, note that our technique actually benefits from safety violations in the human trajectories - the trajectories are not used to guide behaviour, but only to learn about the state space, and safety violations are likely to explore the regions of the state space which are most important to learn about. Conversely, this is not to say that our technique relies on safety violations in the human trajectories - assuming unsafe states are surrounded by a region of near-unsafe states, it should be sufficient for human trajectories to visit these near-unsafe states, as if we can teach the agent to avoid near-unsafe states, the agent should also avoid the unsafe states themselves. To make this clearer, we'll a) be soon adding results of an ablation in which we train the dynamics model only on safe trajectories, showing that performance remains comparable, and b) we'll tweak the text to make the conceptual ideas here clearer, e.g. by explicitly discussing the difference between demonstration trajectories and exploratory trajectories, and being careful to use the latter term rather than the former.
>
> Second, the question of **determining when training is completed**. We expect the ultimate determination here to come from running agent rollouts in the simulated environment - a key advantage our technique has over model-free methods. One difficulty in doing this in practice is that it requires the dynamics model to consistently remain coherent over the entire length of a simulated episode, which ours does not, so this will require some further innovation in generative video models to be practical. In the meantime, there are intermediate signs of model quality we can check. For the dynamics model, model quality can be easily determined by a) checking rollouts with initial states and action sequences sampled from the test set of human trajectories, and b) proving the model interactively, providing actions in real-time to explore the simulated environment. For the reward model, we can a) use validation loss as a coarse-grain indicator of quality (giving us information about whether the reward model is robust enough to predict held-out human feedback), and b) also probe it interactively by exploring the simulated environment and seeing whether the predict rewards behave as expected. Again, we can add a paragraph or two to the paper making all this more explicit.
>
> Third, the question about whether the reward learning setup would be capable of **recovering the sparse reward function** used to train the model-free baseline. Indeed, the feedforward network we use for our main experiments would be unable to recover the sparse reward signal, given that the sparse rewards are based on eating apples, which requires information from multiple timesteps (the apple being present in one timestep and then disappearing in the next timestep). However, we are currently working on an additional experiment adding frame-stacking to the reward model inputs, and this addition should in fact enable the reward model to predict the sparse reward signal. If this is a crux for you in raising your scores, please let us know, and we can prioritise this work over the coming weeks.
>
> Finally, the question of **whether our technique can outperform model-free RL when the reward is well-specified**. We have three responses to this. First, this is early work on ReQueST; our aim at this stage is not to outperform traditional RL (which has had many years to mature), but simply to demonstrate the viability of an alternative. Second, in the majority of real-world tasks we believe it won't be possible to specify perfectly clean rewards using a procedural function, so perhaps model-free RL with a well-specified reward is not the fairest point of comparison; a better comparison might be model-free RL with rewards also predicted by a reward model, and this is certainly a baseline we could add. Third, note that the key metric we care about here is safety, not performance - so again, rather than plain model-free RL, a better comparison might be other safe RL techniques (though see our paper-level comment about the viability of evaluating such baselines for this work).

---

### Author Response · Authors · 2021-11-16
**Reply to all reviewers**

We thank all reviewers for their time reading and understanding our work, and for their thoughtful comments to help us improve.

We address individual reviewers' points in individual comments, but here we'll address one common request: comparison to other safe RL techniques as baselines, such as constrained RL algorithms. Unfortunately, we are not aware of any existing work that a) demonstrates the viability of constrained RL using a constraint model learned from human feedback, and b) has been shown to scale to pixel-based observations. It is unclear whether existing constrained RL techniques can work under both of these assumptions. We might guess that a) can be addressed using a reward model and reward sketching and similar, and b) can be addressed by switching out an MLP for a CNN, but investigating these is outside the scope of our project. If we are mistaken in our beliefs here, please do let us know - it would be extremely useful feedback.

---

### Author Response · Authors · 2021-11-19
**Updated paper revision**

We've uploaded a new revision of the paper that includes:
* Preliminary results using sparse (rather than dense) feedback, providing a comparison to prior work on ReQueST.
* Two additional appendices: F, detailing evaluation of model quality, and G, detailing agent failures.

Thanks!

---

### Decision · Program_Chairs · 2022-01-20

**Decision:**

Reject

**Comment:**

This work takes ReQuest, an approach for safe deep reinforcement learning utilizing human feedback, and studies it's feasibility in pixel-based 3D environments (previously it was only shown to work in simple 2D environments). In order to apply ReQuest in this much more challenging settings, this novel instantiation of ReQuest learns a pixel-based dynamics model from a lot of human demonstration data, and a different (as compared to the "base" ReQuest) reward sketching approach to infer the reward function from human feedback.

**Strengths**
globally a well motivated and well written/structured manuscript
Adresses an important problem, and shows promising results

**Weaknesses**
on the more detailed level, there are some clarity concerns (even with the lengthy appendix)
evaluation was missing some more relevant comparison (partially fixed after rebuttal/revision)
lack of technical novelty, and lack of in depth analysis of results
motivation of algorithmic choices : why did you choose the reward sketching approach that you chose? How is it different, and does it improve performance?

**Rebuttal**
The authors addressed most questions/things that were unclear and updated the paper to include an additional baseline.

Additional baseline: First, it's great that you added this additional baseline! Yet, to me it's unclear what that additional baseline really represents (Request + sparse rewards). The original ReQuest paper also learns reward from human feedback, is that what you did for this paper? If yes then what does the sparse reward mean? Why does this version of request perform worse than your proposed version?

**Summary**
I agree with the reviewers and authors that this is a promising direction. However, in it's current form this manuscript is not ready yet for publication. My  concern are centered around motivation of algorithmic choices: The reward sketching part (while not novel in itself) is a novel component of the ReQuest pipeline, but you do not evaluate what it adds, and neither do you motivate that choice. Furthermore, the additional baseline is not clearly described, it's unclear how it's different for your proposed approach and why we see the performance improvement of your ReQuest version vs the baseline ReQuest version.